# DaVinci: Reinforcing visual-structural Syntax in MLLMs for Generalized Scientific Diagram Parsing

**Xingchen Zeng**[1], **Zhewei Su**[1], **Hengming Zhang**[1], **Juyong Jiang**[1], **Jiazhi Xia**[2], **Wei Zeng**[1]*

[1]The Hong Kong University of Science and Technology (Guangzhou)
[2]Central South University

## Abstract

Parsing raster-based scientific diagrams into structured representations is critical for editability and reusability. However, existing multimodal LLMs (MLLMs) struggle with the diverse visual primitives, complex structural layouts, and strict syntax involved. To address this, we introduce **DaVinci**, a novel MLLM that learns diagram parsing based on a two-stage framework: supervised learning of visual primitives followed by reinforcement learning of their structural relationships. Our model learns visual-structural syntax through supervised training on TikZ30K, a newly curated dataset of high-quality diagram-TikZ code pairs that features abundant visual primitives and structurally optimized drawing sequences. We further refine the model via reinforcement learning, guided by a hybrid reward function that jointly optimizes for visual fidelity, structural consistency, and code correctness. Extensive experiments show that **DaVinci** significantly outperforms existing open-source MLLMs and surpasses leading proprietary models like GPT-5 and Claude-Sonnet-4. Code, datasets, and models are available[1].

## 1 Introduction

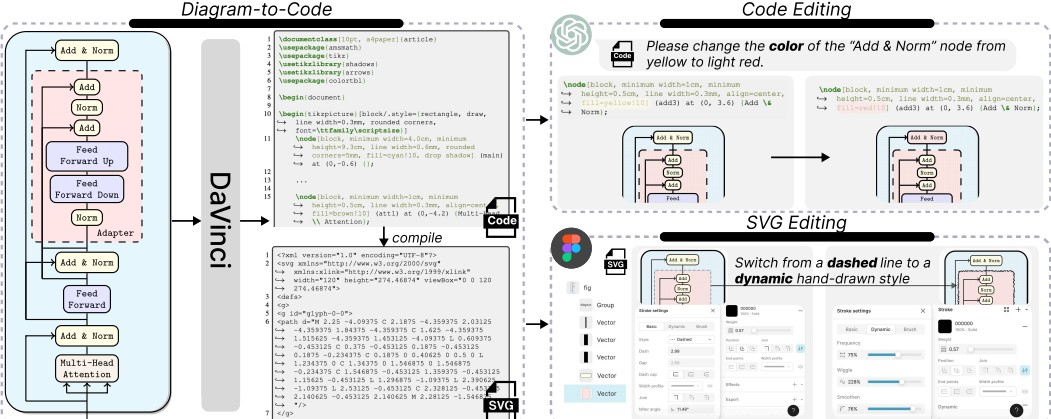

Figure 1: DaVinci parses scientific diagram images into structured TikZ code, which can subsequently be compiled into SVG (Left). The TikZ code enables LLMs to edit diagrams through code manipulation; SVG enables individual element editing using illustration software like Figma (Right).

Scientific diagrams are ubiquitous across research, engineering, and education, serving as essential visual tools for conveying complex concepts, relationships, and events (Kembhavi et al., 2016; Hu et al., 2024; Zeng et al., 2025). Automatically parsing rasterized diagrams into structured, programmatic representations is a crucial step toward unlocking their editability, reusability, and integration into downstream workflows (as illustrated in Figure 1), offering an alternative to time-consuming manual reconstruction (Belouadi et al., 2024b).

---

*Wei Zeng is the corresponding author. E-mail: wei-zeng@hkust-gz.edu.cn.

[1]https://github.com/zengxingchen/Diagram-to-TikZCode

Among common diagram representations, Ti*k*Z is particularly well-suited for diagram-to-code generation in the context of multimodal large language models (MLLMs). Ti*k*Z features declarative syntax that is mathematically expressive, closely mirrors the logical structure of scientific content, and remains both concise and semantically rich. These features have made it widely employed in recent works (Belouadi et al., 2024b;a; 2025; Wang et al., 2025).

Existing works have collected large-scale diagram-Ti*k*Z code datasets to support supervised fine-tuning (SFT) for improving MLLMs' diagram parsing capabilities. For example, Belouadi et al. (2024a;b; 2025) collect DATı*k*Z series data features diagram-Ti*k*Z code pairs from real-world sources such as arXiv papers and curated GitHub repositories. Detikzify (Belouadi et al., 2024b) further introduces an MCTS-based inference algorithm to improve the perceptual similarity (Zhang et al., 2018) of the output image. Despite the progress, current MLLMs still struggle with this task, primarily because parsing diagrams demands pixel-level precision in encoding geometric details. This requires the exact specification of diverse graphical primitives, such as lines, shapes, and text, along with their precise spatial relationships, all while adhering to TikZ's strict syntactic constraints.

In this paper, we introduce DaVinci, a novel MLLM that learns diagram parsing based on a two-stage framework: supervised learning of visual primitives, followed by reinforcement learning of their structural relationships. Specifically, we identified two significant yet underexplored data features for diagram parsing: drawing order and comment annotations. We extend the DATı*k*Z series data collection pipeline with enhanced quality control and data enhancement.

Following initialization with SFT, we employ Group Relative Policy Optimization (GRPO) (Shao et al., 2024) to refine the model's ability to generate runnable and visually faithful Ti*k*Z codes from input diagram images. To construct accurate and explicit rewarding signals, we design a hybrid reward that constructs reward signals from the generated code, vectorized representation, and rendered image. Specifically, we innovatively employ the vectorized representation to extract textual and geometric primitives in an extraction-error-free manner to provide precise feedback on the generated diagrams' spatial-textual alignment and geometric accuracy. Experimental results show that DaVinci significantly outperforms existing open-source MLLMs and even surpasses leading commercial models (e.g., GPT-5, and Claude-Sonnet-4) in diagram parsing.

Our main contributions are summarized as follows:

- We introduce DaVinci, a novel MLLM that learns diagram parsing based on a two-stage framework: supervised learning of visual primitives followed by reinforcement learning of their structural relationships, which demonstrates superior performance.
- We identify the significance of drawing order and comment annotations for diagram parsing, and construct a high-quality diagram-code dataset Ti*k*Z-30K.
- We design a novel Hybrid Reward function for reinforcement learning, leveraging the vectorized representation to construct spatio-textual and geometric rewards in an extraction-error-free manner, mitigating the errors introduced by OCR and other heuristic parsers.

## 2 Related Work

**Image-to-Code Generation** aims to reproduce visual inputs with editable and structurally faithful code (Jiang et al., 2025b). This growing field is exemplified by efforts targeting user interfaces (UI) (Si et al., 2024; Yun et al., 2024), charts (Li et al., 2025a; Zhao et al., 2025), and diagrams (Belouadi et al., 2024b; Wei et al., 2025), each posing unique semantic and structural challenges. For instance, *chart-to-code* operates over closed-set chart types and their constrained grammar spaces within plotting libraries such as Matplotlib (Shi et al., 2025). In contrast, UI and diagram code generation involve open-ended composition over diverse visual primitives. Specifically, the diagram code encodes geometric details with pixel-level precision, requiring the exact specification of diverse primitives (e.g., lines, shapes, text) and their spatial relationships (Belouadi et al., 2024b). To tackle this challenge, our work employs reinforcement learning with complementary feedback from image, vectorized representation, and codes for visual fidelity, geometric accuracy, and code correctness.

**Diagram Generation.** Diagram code spans several widely used formats, including Ti*k*Z, SVG, and template-based libraries (Mermaid, 2025). SVG provides the most primitive representation among these formats, but offers limited semantic clarity. Moreover, existing research on SVG generation has primarily targeted icons and anime characters (Li et al., 2025b; Yang et al., 2025), likely due to

the complexity of scientific diagrams and the scarcity of high-quality SVG diagram datasets (Rodriguez et al., 2025a). In contrast, TikZ strikes a balance between semantic clarity and visual expressiveness, with native support for mathematical notation and seamless export to vector formats such as PDF and SVG. DATikZ series (Belouadi et al., 2024a;b; 2025) collects large-scale TikZ programs from real-world sources like arXiv papers, establishing foundational resources for diagram generation (Wang et al., 2025). Our work builds upon their collection methods and further introduces enhanced quality control and key data enhancement specific to diagram generation problems, including drawing order and leveraging comments as planning scaffolds.

**Reinforcement Learning Post-Training** has demonstrated remarkable success in refining the reasoning capabilities and alignment of LLMs (Chu et al., 2025; Chen et al., 2025). Among RL approaches, rule-based methods like GRPO (Guo et al., 2025) have gained prominence due to their simplicity and effectiveness. This paradigm has been extended to visual content generation, where reward signals can be derived from the generated images (Jiang et al., 2025a; Xue et al., 2025). Specifically, Rodriguez et al. (2025b) introduces rewards based on pixel-level L2 loss, DreamSim-based feature similarity (Fu et al., 2023), and code compactness for SVG generation. ChartMaster (Tan et al., 2025) evaluates the visual fidelity of the generated charts using CNN-based features together with key visual attributes extracted directly from the code. However, these reward signals are limited in their accuracy in capturing the spatial-textual alignment and geometric features of diagrams. We introduce extraction-error-free spatio-textual and geometric rewards derived directly from visual primitives extracted from the vectorized representation of diagrams.

## 3 DaVinci

### 3.1 Overview

We formulate the diagram parsing as a conditional sequence generation task in the MLLM context. Given an input image $I_{in}$, the goal is to generate a TikZ code sequence $C_{pred} = (t_1, \ldots, t_L)$ that can be rendered into an image $I_{pred}$ that faithfully reconstructs $I_{in}$. Formally, we employ an MLLM parameterized by $\theta$ to model the conditional distribution $p(C_{pred}|I_{in})$. The optimal code sequence $C_{pred}^*$ is obtained via autoregressive decoding:

$$C_{pred}^* = \arg\max_{C_{pred} \in \mathcal{T}} p(C_{pred}|I_{in}; \theta) = \arg\max_{C_{pred} \in \mathcal{T}} \prod_{i=1}^{L} p(t_i|t_{<i}, I_{in}; \theta)$$

Here, $\mathcal{T}$ represents the space of valid TikZ drawing commands. The fidelity of the rendered image measures the ultimate success of the generation, $I_{pred} = \text{Render}(C_{pred}^*)$, to the original input $I_{in}$.

We trained DaVinci with a two-stage framework: 1) cold-start initialization with supervised learning of visual primitives and grammar rules with our curated TikZ30K (Sect. 3.2); followed by 2) reinforcement learning for refining output diagrams' spatial-textual alignment, geometric accuracy, overall visual fidelity, and code correctness (Sect. 3.3).

### 3.2 Dataset Construction for Cold-start Initialization

TikZ-30K distinguishes itself through three fundamental considerations that enable effective cold-start: (1) *high-quality images with semantic distribution*, (2) *drawing order normalization*, and (3) *comment-rich annotations*. We provide an overview here and more details in Appendix C.

**Data Collection.** Our data pipeline begins with reproducing the collection process of the DATikZ series (Belouadi et al., 2024a;b; 2025), since the non-distributable licenses of some underlying data sources prevent the public release of their complete data. To ensure strict temporal separation from the DATikZ$_{v3}$ test set, which includes data from January 2024 onward, our training data is restricted to sources published by December 2023, thereby eliminating the risk of contamination. We initially source 366,075 runnable TikZ snippets from TeX.SE, arXiv papers, and curated GitHub repositories, rendering them into images for downstream processing. However, the collected data still contains substantial noise regarding code syntax and visual content, and lacks labels for their categorical distribution, leaving the semantic distribution of TikZ programs unclear.

**Data Filtering.** Through manual inspection, we identify recurring patterns of low-quality samples to derive a set of heuristic filtering rules, such as truncated images due to LaTeX overfull errors, where

content extends beyond the defined page margins and is consequently clipped during compilation. The rule-based filtering reduces the dataset to 258,421 samples. We then employ Qwen-2.5-VL-32B (Bai et al., 2025) to automatically assign semantic class labels and 5-point quality scores to each image, retaining only those with scores of 4 or 5, yielding final 225,648 high-quality samples. Full details of the data distribution and filtering criteria are provided in Appendix C.

Notably, we identify two underexplored yet critical features to diagram generation: *code ordering* and *descriptive comments*, which directly impact model training effectiveness.

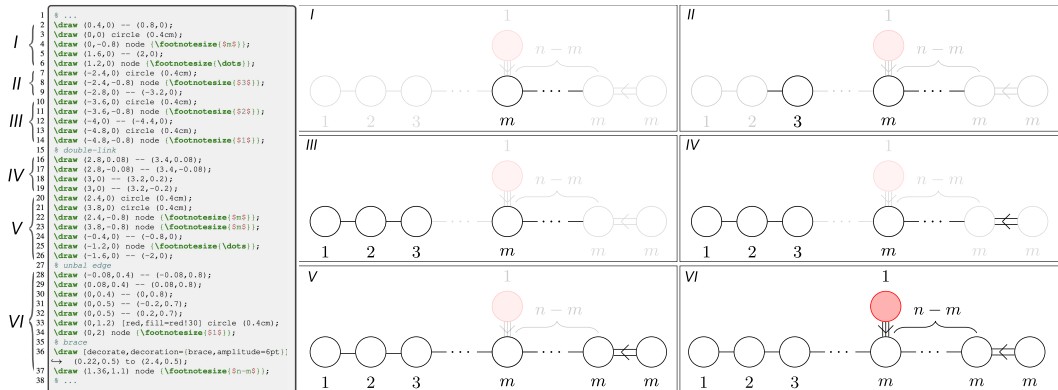

Figure 2: Drawing order visualization of a collected sequential diagram. (Left) Original code labeling six blocks (I-VI). (Right) Block-by-block rendering results, revealing the noisy drawing order.

**Code Reordering.** In general-purpose programming languages like Python, code order is tightly constrained by syntax and execution dependencies. However, the rendering of drawing language like SVG and Ti*k*Z is largely independent of code ordering, except for variable declaration, presenting special data noises. For example, Figure 2 illustrates the ordering noise with the block-by-block rendering of a sequential diagram in our collected data. The original code drawing begins with the center node $m$ along with its outgoing edge to the right (I). It then abruptly shifts to render node 3, followed by nodes 2 and 1 in right-to-left order (II, III). Notably, for the two rightmost $m$ nodes, intermediate edges are drawn *before* the nodes themselves (IV, V). The noisy drawing order does not affect the rendering result but is detrimental to the autoregressive language model. Specifically, different training samples may map similar visual contents (e.g., a graph layout) to multiple, arbitrarily permuted code sequences, largely degrading the training effectiveness.

To address the ordering issue, we employ Qwen3-Coder-480B-A35B-Instruct (Qwen Team, 2025) to reorder the Ti*k*Z codes, following a semantics-guided, logically constructive drawing protocol. Figure 11 presents the optimized code and block-by-block rendering result, aligning with human intuition and logical process. Post-verification is conducted to ensure the consistency of the rendering results before and after reordering.

**Injecting Comments as a Planning Scaffold.** Raw Ti*k*Z code is often a dense sequence of low-level drawing commands, lacking explicit structural or semantic cues except for connection relationships. While some collected codes contain comment annotations, they are often incomplete and lack consistency. This poses a significant challenge for models' global planning capability over complex diagrams. Without such scaffolding, the model is prone to logical inconsistencies and omitted components, especially for long code sequences.

To mitigate this, we leverage LLMs to systematically enrich the training data with comments that decompose the drawing process into semantically meaningful sub-tasks. For instance, in the example in Figure 11, comments such as *Start from the far left: draw the first circle, label it "1", and connect it rightwards* or *Add a brace annotation to visually indicate that the rightward extension covers n-m nodes* explicitly state the model's current drawing intent. These comments serve as planning anchors that guide the model to complete the drawing process step by step.

To enable efficient cold-start training while preserving representativeness, we perform stratified sampling by token length over the 225,648 filtered samples, yielding a balanced subset of 58,000 samples. This subset is split into two parts: 30,000 for SFT and 28,000 for RL. Of the 30,000 SFT samples, 29,859 passed post-verification after code augmentation, denoted Ti*k*Z30K.

## 3.3 Rewarding Design for Diagram Parsing

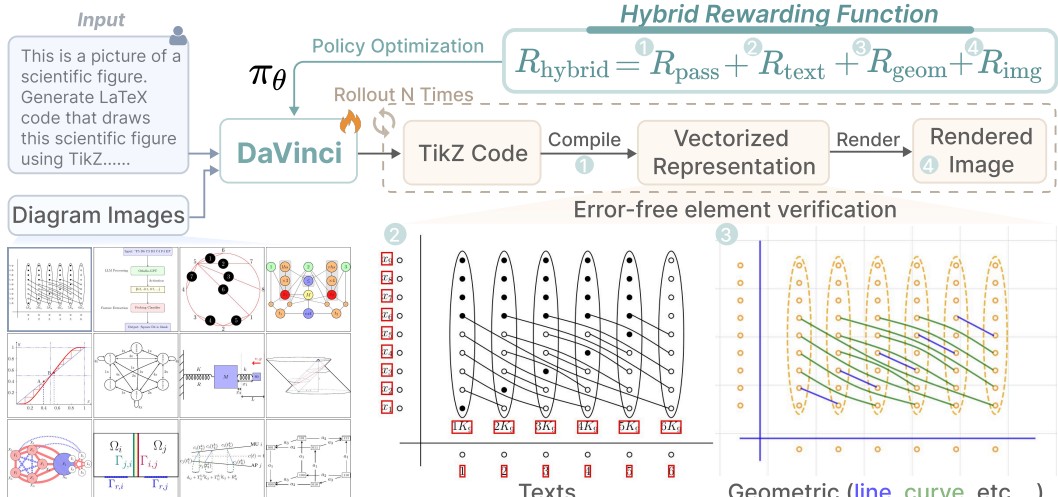

Figure 3: The reinforcement learning framework of DaVinci. We construct complementarity reward signals from the generated code, vectorized representation, and rendered image. Specifically, we extract the textual and geometric elements from the vectorized representation in an error-free manner.

Following SFT, we apply reinforcement learning to refine the LLM's ability to generate runnable and visually faithful Ti*k*Z codes from input diagram images. Specifically, we employ Group Relative Policy Optimization (GRPO) (Shao et al., 2024), a variant of Proximal Policy Optimization (PPO) (Schulman et al., 2017) that estimates advantages through group-relative comparisons, eliminating the need for an explicit critic model.

Given a prompt $T$, the current policy $\pi_\theta$ (initialized from the model after SFT) generates a set of $G$ diverse candidate sequences $\{A_k\}_{k=1}^G$, where each $A_k$ corresponds to the Ti*k*Z code $C_k$. For each candidate, we compute a hybrid reward $R_{\text{hybrid}}^{(k)}$ and derive its advantage $\hat{A}_k$ relative to the group's empirical performance:

$$\hat{A}_k = \frac{R_{\text{hybrid}}^{(k)} - \text{mean}\big(\{R_{\text{hybrid}}^{(j)}\}_{j=1}^G\big)}{\text{std}\big(\{R_{\text{hybrid}}^{(j)}\}_{j=1}^G\big)}. \tag{1}$$

The computed advantage is used to update the policy following the GRPO objective, detailed in Appendix B. To effectively guide the RL training, we design a novel hybrid reward function specifically designed for diagram generation. Specifically, we construct signals from *image $I$*, *vectorization representation $V$*, *code $C$*, three different modalities. For each generated candidate $C_k$ from prompt $T_k$, the total reward $R_{\text{hybrid}}^{(k)}$ is defined as a weighted sum of four complementary components:

$$R_{\text{hybrid}} = \underbrace{R_{\text{text}}(V_{\text{in}}, V_{\text{pred}})}_{\text{Spatio-Textual Reward}} + \underbrace{R_{\text{geom}}(V_{\text{in}}, V_{\text{pred}})}_{\text{Geometric Reward}} + \underbrace{R_{\text{img}}(I_{\text{in}}, I_{\text{pred}})}_{\text{Image Fidelity}} + \underbrace{R_{\text{pass}}(C_{\text{pred}})}_{\text{Compile Success}}, \tag{2}$$

where we do not set special weights for each reward component. Diagrams mainly consist of textual and geometric elements; thus, we separately design explicit and effective signals for both of them. The individual reward components are defined as follows.

**Spatio-Textual Similarity Reward** $R_{\text{text}}$. Optical Character Recognition (OCR) is commonly used to extract text contents and their corresponding bounding boxes from both images. However, OCR is error-prone for diagrams that contain diverse symbols and involve easily misrecognized graphical elements, and thus may propagate errors to the final rewarding signal. Appendix E.4 provides common failure cases we found when using the state-of-the-art OCR models.

To address this, we leverage the PDF-format vectorization representation generated by TikZ to extract the text content $(T_{pred}, T_{gt})$ and their corresponding bounding boxes $(B_{pred}, B_{gt})$ in a ***error-free*** manner. Unlike rasterized images that rely on visual recognition for text extraction, PDFs

generated by TikZ retain the exact geometric and typographic metadata of every text object as native vector elements. We exploit this property using the PyMuPDF (2024) library to directly access the text objects that contain the ground-truth text characters and corresponding bounding boxes. We then employ a two-step exact-then-vaguely approach. First, it greedily matches elements with identical recognized text. Second, for the remaining elements, it uses the Levenshtein distance with an adaptive threshold to pair texts with minor OCR errors. In any case where one-to-many matches occur, the conflict is resolved by selecting the pair with the highest Distance-IoU ($d_{IoU}$) (Zheng et al., 2020), ensuring geometric plausibility. Finally, the total reward $R$ is calculated from all successful matches from both stages. The $d_{IoU}$ of each matched pair, originally in the range $[-1, 1]$, is transformed into a score of $[0, 1]$. These scores are summed and then normalized by the maximum number of elements in either the predicted or ground-truth set, yielding a comprehensive and bounded reward score:

$$R_{\text{text}} = \frac{1}{\max(|B_{pred}|, |B_{gt}|)} \sum_{(b_p, b_g) \in \text{Matches}} \frac{d_{IoU}(b_p, b_g) + 1}{2} \tag{3}$$

The full procedure is outlined in Algorithm 1. Figure 17 presents qualitative examples of the Spatio-Textual match results.

**Geometric Similarity Reward** $R_{\text{geom}}$. Directly comparing rendered images using pixel-wise metrics (e.g., Mean Squared Error or SSIM (Wang et al., 2004)) is inadequate to capture the underlying structural correspondence of geometric primitives.

To overcome these limitations, $R_{\text{geom}}$ assesses structural fidelity by directly comparing geometric elements. First, similar to the processing of textual elements, we use the PyMuPDF (2024) library to access the geometric primitives ($E_{pred}, E_{gt}$) from predicted and ground-truth PDFs, such as lines, rectangles, and circles, along with their attributes. Second, for each geometric type, we formulate a bipartite matching problem to establish the optimal one-to-one correspondence between predicted and ground-truth elements. The matching is solved using the Hungarian algorithm, guided by a type-specific cost function $C(e_p, e_g)$ that measures the dissimilarity between a pair of elements. This cost is a weighted sum of differences in key geometric attributes, including the normalized centroid distance, the relative size (area or length), and the orientation or aspect ratio.

Finally, the total reward is computed by aggregating the quality of all successful matches. The cost of each matched pair is transformed into a similarity score via an exponential decay function, rewarding pairs with lower geometric dissimilarity. These scores are then summed and normalized by the maximum number of elements in either set to ensure the reward is bounded within $[0, 1]$. This is formally defined as:

$$R_{\text{geom}} = \frac{1}{\max(|E_{pred}|, |E_{gt}|)} \sum_{(e_p, e_g) \in \text{Matches}} \exp(-k \cdot C(e_p, e_g)) \tag{4}$$

where Matches is the set of optimal pairings, $C(e_p, e_g)$ is the cost function for a pair of elements, and $k$ is a scaling constant. The complete procedure is detailed in Algorithm 2. Figure 18 presents qualitative examples of the geometric element match results.

**Image Fidelity Reward** $R_{\text{img}}$. We combine DreamSim (Fu et al., 2023) and Mean Squared Error (MSE) for measuring image fidelity in both feature and pixel space. DreamSim is a perceptual metric trained on human judgment data, making it well-suited for assessing overall visual similarity for diagrams. MSE is simple but effective in capturing absolute details.

$$R_{\text{img}} = \text{DreamSim}(I_{\text{in}}, I_{\text{pred}}) + \text{clip}\left(1 - \frac{1}{s^2} \left\| I_{\text{in}}^{\text{norm}} - I_{\text{pred}}^{\text{norm}} \right\|_2^2, -1, 1\right), \tag{5}$$

where $I^{\text{norm}} = (I - \mu)/\sigma$ represents the normalized image, and $\text{clip}(x, a, b) = \min(\max(x, a), b)$ bounds the MSE signal within $[-1, 1]$.

**Compile Success Reward** $R_{\text{pass}}$. We do not explicitly integrate a bonus value for compilation success. Instead, if a generated TikZ program fails to compile, we assign the minimum possible value to the other three reward components. This design ensures that non-runnable code receives the lowest possible hybrid reward.

## 4 Experiments

### 4.1 Experimental Setup

To rigorously evaluate the performance of various vision-language models on Image-to-Ti*k*Z translation tasks, we conduct a series of controlled experiments comparing both proprietary and open-weight models. The evaluation focuses on the models' ability to generate syntactically correct, semantically faithful, and visually accurate Ti*k*Z code from input diagrams.

**Cold start with supervised fine-tuning.** We train the base model Qwen2.5-VL-7B-Instruct (Bai et al., 2025) on the TikZ-30K dataset for 2 epochs, yielding DaVinci-SFT-7B.

**RL Post-training.** We initialize the policy model with DaVinci-SFT-7B. We use the GRPO algorithm with a global batch size of 512 and a rollout batch size of 256. We use a rollout number of 10. The training is performed on 8 * H100-80G for 500 steps, yielding DaVinci-7B. Detailed training settings are in Appendix E.3.

### 4.2 Automatic Evaluation

**Baselines.** We compare *DaVinci* against a diverse set of baselines, including:1) *Proprietary MLLMs:* Gemini-2.5-Pro (Comanici et al., 2025), GPT-5-Default (self-control the thinking efforts) (OpenAI, 2025), and Claude-Sonnet-4, Claude-Sonnet-4-Thinking (Anthropic, 2025); 2) *Open Sourced MLLMs:* Qwen2.5-VL (Bai et al., 2025) and GLM4.5V series (Team, 2025); 3) *Specialized TikZ-generation models:* DetikZify-V2-8B (Belouadi et al., 2025), DiagramAgent-7B (Wei et al., 2025).

**Benchmarks.** We report results on the official test set of $\text{DAT}ikZ_{v3}$ (Belouadi et al., 2025), 542 visually complex and diverse graphics selected from the whole dataset.

**Metrics.** We evaluate performance at both code and image levels. At the code level, we report the Pass@1 compile rate, Text Edit Distance (TED), and CrystalBLEU (cBLEU) (Eghbali & Pradel, 2022). At the image level, we assess visual fidelity using DreamSim (Fu et al., 2023), SigLIP (Zhai et al., 2023), Structural Similarity Index Measure (SSIM) (Wang et al., 2004), Mean Squared Error (MSE), and LPIPS (Zhang et al., 2018). Additional metrics evaluating code efficiency are provided in Table 7 due to space limitations.

Table 1: Evaluation results of DaVinci against different models. **Bold** and underlined values denote the best and second-best scores. Arrows ↑↓ indicate metric directionality.

| Model | Code | | | Image | | | | |
|---|---|---|---|---|---|---|---|---|
| | Pass@1↑ | TED↓ | cBLEU↑ | DSIM↑ | SigLIP↑ | SSIM↑ | MSE↓ | LPIPS↓ |
| *Proprietary MLLMs* | | | | | | | | |
| Gemini-2.5-Pro-Thinking | 69.93 | **53.77** | 6.17 | **88.20** | **95.59** | **75.86** | 66.62 | **21.64** |
| GPT-5-Default | 72.88 | 53.17 | 3.22 | 83.78 | 93.79 | 73.73 | 73.57 | 25.37 |
| Claude-Sonnet-4 | 84.87 | 54.33 | 3.90 | 83.81 | 92.64 | 73.45 | 72.69 | 24.93 |
| Claude-Sonnet-4-Thinking | 86.90 | 54.38 | 3.31 | 82.89 | 92.87 | 72.21 | 73.89 | 25.80 |
| *Open Sourced MLLMs* | | | | | | | | |
| Qwen2.5-VL-7B | 59.59 | 57.35 | 5.28 | 74.37 | 89.12 | 71.61 | 82.55 | 29.00 |
| Qwen2.5-VL-32B | 76.20 | 54.80 | 4.71 | 76.17 | 89.12 | 73.73 | 78.36 | 27.28 |
| Qwen2.5-VL-72B | 80.04 | 54.80 | 4.18 | 79.35 | 91.52 | 72.42 | 77.63 | 27.24 |
| GLM-4.5V-106B-A12B | 67.90 | 54.72 | 3.42 | 79.33 | 91.36 | 73.87 | 77.52 | 26.37 |
| GLM-4.5V-Thinking | 62.92 | 54.75 | 3.08 | 78.05 | 90.77 | 72.59 | 78.71 | 27.53 |
| *Specialized TikZ-generation MLLMs* | | | | | | | | |
| DetikZify-V2-8B | 78.60 | 55.66 | 7.19 | 82.63 | 91.99 | 74.30 | 68.42 | 23.30 |
| DiagramAgent-7B | 57.75 | 56.85 | 5.55 | 70.60 | 87.39 | 72.30 | 84.34 | 29.28 |
| DaVinci-SFT-7B | 84.50 | 56.21 | **7.52** | 81.15 | 92.27 | 72.65 | 73.90 | 26.10 |
| DaVinci-7B | **97.60** | 55.13 | 6.57 | 84.83 | 93.93 | 73.65 | **61.81** | 22.32 |

### 4.3 Result Analysis

DaVinci-7B overtakes the best scores across all open-sourced models, and surpasses the leading commercial models like GPT-5 and Claude-Sonnet-4. Gemini-2.5-Pro presents better performance

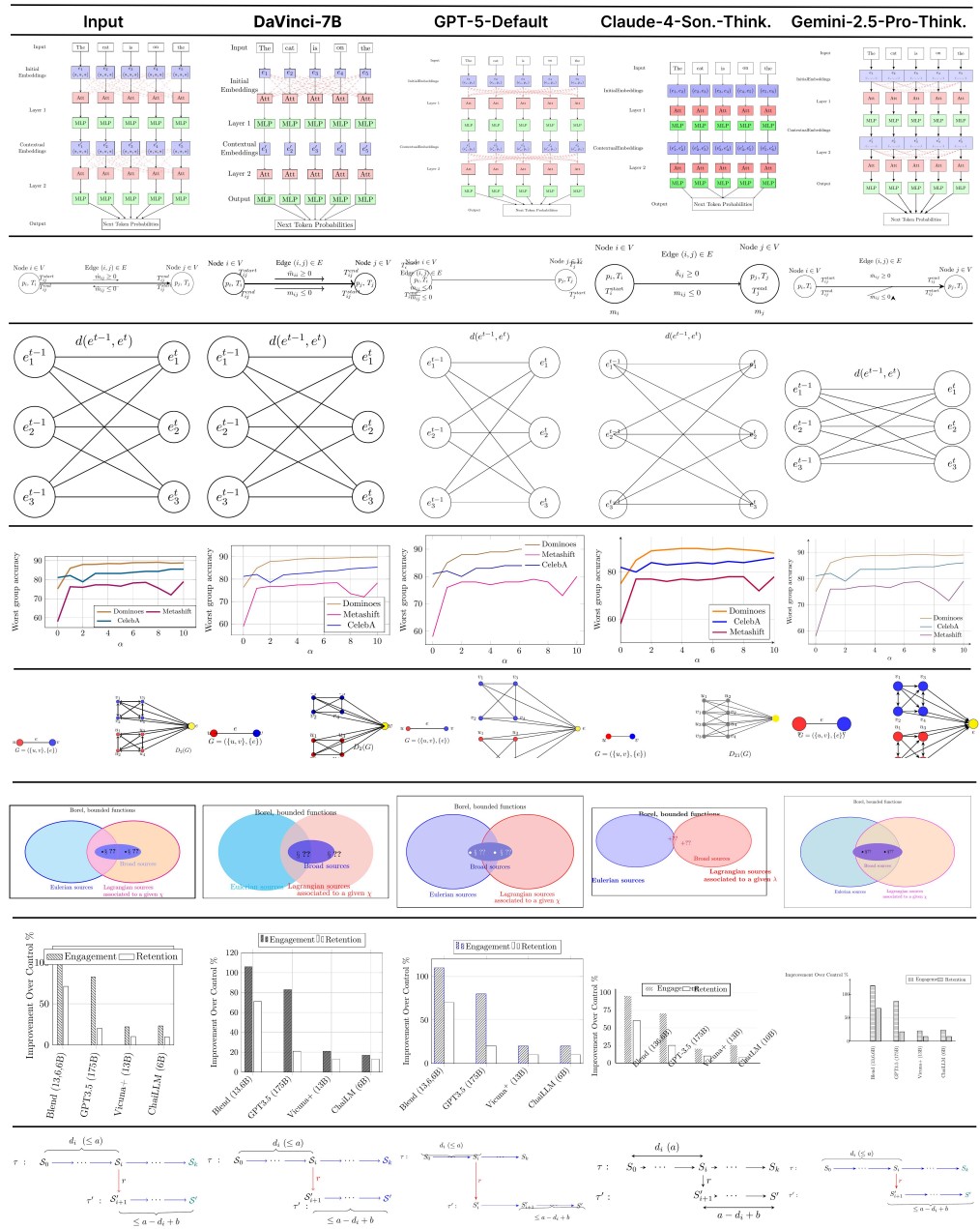

Figure 4: Examples of model inputs and generated outputs from DaVinci-7B and the advanced commercial models.

than DaVinci-7B regarding certain metrics such as DreamSim and LIPIS, but with a significant gap in compile rate. We detail below the analysis of the results.

**Compile Rate.** DaVinci-SFT-7B achieves the third-highest compile rate (84.50%), slightly below Claude-Sonnet-4-Thinking (86.90%). Since the base model of DaVinci-SFT-7B is stronger than DetikZify-V2-8B (78.60%), we conduct an isolated ablation study in Sect. 4.5 to investigate the impact of our data strategy on compile rate improvements. After RL post-training, DaVinci-7B attains a near-perfect compile rate (97.60%). The few remaining failure cases are mainly dense visualizations like scatter plots, where the model over-produces data points, leading the output to exceed the context limit. In contrast, Gemini-2.5-Pro exhibits a notably lower compile rate (69.93%) despite its strong performance on other metrics. Inspection of its compilation logs reveals frequent omissions of essential LaTeX library imports (e.g., `decorations`, `groupplots`, `sloped`, `shapes`) and missing basic symbols (e.g., parentheses, math delimiters), which prevent successful rendering.

**High Code Similarity Is Not Necessary.** We find that specialized SFT models, DaVinci-SFT-7B and DetikZify-V2-8B, achieve the highest cBLEU scores, indicating greater lexical similarity to reference code. However, after RL training, DaVinci-7B's cBLEU score decreases while all other metrics improve, including visual fidelity and compile rate. This observation underscores a key insight: visually equivalent outputs can be produced by syntactically diverse TikZ code. Thus, strict code-level similarity is neither necessary nor always desirable for high-quality diagram parsing.

**To Think or Not to Think (Jiang et al., 2025c).** Unlike models that generate explicit reasoning traces before code output, DaVinci-7B uses inline code comments as lightweight planning scaffolds rather than separate reasoning steps. To assess the impact of explicit thinking, we compare thinking-enabled and non-thinking variants of models that support both modes, *i.e.*, Claude-Sonnet-4 and GLM-4.5V. Gemini-2.5-Pro and GPT-5 are only available in their thinking variants. Surprisingly, we find that enabling explicit reasoning does not consistently improve diagram parsing performance. In fact, GLM-4.5V-Thinking shows a significant drop in compile rate (62.92%) compared to its non-thinking counterpart (67.90%). This suggests that for structured code generation tasks like diagram parsing, the act of producing code itself may serve as an implicit reasoning process, with each drawing command attending to specific visual elements in the input. We leave a deeper investigation of this phenomenon to future work.

## 4.4 Human Evaluation

**Settings.** We perform human evaluation studies using *Best-Worst Scaling* (BWS) (Flynn & Marley, 2014), evaluating 100 items randomly sampled from the test set, in line with prior work (Belouadi et al., 2024b; 2025). For each reference figure, four output images generated by different models are displayed side-by-side; see Figure 16 for the annotation interface. Six human evaluators, graduate students aged 23 to 29 years ($\mu$=25.83, $std$=2.23; 2 male, 4 female), are tasked to select the single best and single worst output from each set. Each evaluator is compensated with approximately US$15 for their time. These pairwise selections are aggregated and converted into normalized scores ranging from -1 (worst) to 1 (best), computed as the difference between the proportion of times a model's output is chosen as best ($p_{best}$) and the proportion chosen as worst ($p_{worst}$). Specifically, we conduct two groups of human evaluations to ensure comprehensiveness: 1) DaVinci-7B and the top three non-proprietary models according to automatic metric evaluation, *i.e.*, DetikZify-V2-8B, Qwen2.5-VL-72B, and DaVinci-SFT-7B; 2) DaVinci-7B and three proprietary models, *i.e.*, Gemini2.5-Pro-Thinking, GPT5-Default, and Claude-Sonnet-4-Thinking.

**Results.** Table 2 and 3 present the two groups of human evaluation results, which generally align with the automatic evaluation metrics. In the non-proprietary model group, DaVinci-7B ($\mu$=0.365) achieves the highest score, outperforming all models. DaVinci-SFT-7B and Detikzify-V2-8B exhibit comparable performance, while Qwen2.5-VL-72B performs the worst, highlighting the potential improvement by scaling TikZ-specialized models. Consistent with automatic evaluation, Gemini-2.5-Pro-Thinking ($\mu$=0.50) significantly outperforms all other models in both groups, showing its superior capability in this task. Notably, DaVinci-7B also demonstrates stronger performance than GPT-5-Default and Claude-Sonnet-4-thinking in terms of $p_{best}$ and $p_{worst}$. We further measure the agreement among annotators using split-half reliability (SHR) (Flynn & Marley, 2014). The resulting SHR values ($\rho_{\text{Group 1}}$=0.7227 and $\rho_{\text{Group 2}}$=0.7878) indicate strong inter-annotator agreement, supporting the reliability of our human evaluation results; see Appendix D.4 for calculation details.

| Model | $p_{\text{best}}$ | $p_{\text{worst}}$ | score | std |
|---|---|---|---|---|
| Qwen2.5-VL-72B | 0.13 | 0.39 | -0.26 | 0.06 |
| DetikZify-V2-8B | 0.25 | 0.30 | -0.05 | 0.06 |
| DaVinci-SFT-7B | 0.16 | 0.21 | -0.05 | 0.04 |
| DaVinci-7B | **0.47** | **0.11** | **0.36** | **0.03** |

Table 2: Human evaluation results of Group 1.

| Model | $p_{\text{best}}$ | $p_{\text{worst}}$ | score | std |
|---|---|---|---|---|
| Gemini-2.5-Pro-Thinking | **0.58** | **0.08** | **0.50** | 0.10 |
| GPT-5-Default | 0.13 | 0.26 | -0.13 | **0.03** |
| Claude-Sonnet-4-Thinking | 0.10 | 0.45 | -0.35 | 0.05 |
| DaVinci-7B | 0.20 | 0.21 | -0.01 | 0.06 |

Table 3: Human evaluation results of Group 2.

## 4.5 Ablation Study

**Code Reordering and Comments Injection.** We first validate the effectiveness of code reordering and comment injection. We perform a full-parameter SFT on the base model Qwen2.5-VL-7B-Instruct for two epochs with different data settings: 1) Original30K: the original codes of TikZ-30K

that are not optimized regarding ordering and comment annotations; 2) Reordering30K: remove comments in TikZ-30K while the optimized code ordering remains; 3) TikZ-30K. The training settings are the same as the cold start stage.

Our results show that reordering alone increases the compilation success rate by 9.04%↑ over the baseline (Original30K), and additional comment annotations provide a further gain of 5.72%↑. These findings underscore the critical role of both structural reordering and semantic annotations in diagram parsing.

| Model | Pass@1 ↑ |
|---|---|
| Baseline: Qwen2.5-VL-7B | 59.59 |
| + Original30K | 69.74 |
| + Reordering30K | 78.78 |
| + TikZ30K | **84.50** |

Table 4: Comparison of Pass@1(%) across different SFT datasets settings.

**Ablation of Rewards.** We validate the impact of different reward settings based on DaVinci-SFT-7B. The baseline is combining MSE and DSIM rewards. The results are shown in Table 5, showcasing the effectiveness of each reward component.

Table 5: Comparison of different rewarding settings.

| Reward(s) | Image | | | | | | |
|---|---|---|---|---|---|---|---|
| | DSIM ↑ | SigLIP ↑ | SSIM ↑ | MSE ↓ | LPIPS ↓ | Texual ↑ | Geometry ↑ |
| Base ($R_{\text{img}} + R_{\text{pass}}$) | **85.00** | 93.67 | 73.07 | 64.58 | 22.94 | 37.23 | 41.44 |
| Base + SSIM | 84.67 | 93.47 | 73.58 | 64.89 | 23.03 | 37.45 | 41.87 |
| Base + $R_{\text{text}}$ | 84.85 | 93.91 | 73.61 | 63.35 | 22.44 | 41.58 | 42.12 |
| Base + $R_{\text{text}} + R_{\text{geom}}$ | 84.75 | **93.93** | **74.01** | **62.30** | **22.32** | **42.28** | **44.10** |

## 4.6 Case Studies

Figure 4 presents qualitative comparisons between DaVinci-7B and leading commercial models. DaVinci-7B demonstrates superior precision in text placement, geometric alignment, and structural coherence. Additional visual results are provided in Appendix D.2.

## 5 Conclusion

In this work, we introduced **_DaVinci_**, a novel multimodal large language model that advances scientific diagram parsing through a two-stage framework combining supervised learning of visual primitives and reinforcement learning of structural syntax. We curated the high-quality TikZ30K dataset, featuring optimized drawing order and comments annotations as planning anchors to enable MLLMs to learn the visual-structural syntax of scientific diagrams more effectively. For reinforcement learning, we designed a hybrid reward function that leverages vectorized representations to provide signals for geometric accuracy and textual alignment during reinforcement learning. Together, these innovations enable _DaVinci_ to achieve state-of-the-art performance, outperforming both open-source MLLMs and leading proprietary models such as GPT-5 and Claude-Sonnet-4.

## Data Release and License Information

Our data release strictly adheres to the licensing terms of the original sources from which we extract the TikZ code snippets. For code snippets sourced from permissively licensed platforms that allow redistribution and adaptation, we directly release the original and optimized code, including the Creative Commons Attribution Licenses[2], the GNU Free Documentation License[3], or the MIT license[4] for websites, GitHub repositories, and part of arXiv sources, with clear data source attribution per code snippet. For the data where the original license limits redistribution (_i.e._, the arXiv.org perpetual, non-exclusive license[5]), we do not redistribute the raw source. Instead, we provide diff files and reproducible scripts that enable researchers to locally reconstruct the optimized code in full compliance with their licensing terms, without extra time or financial costs for code optimization. This approach ensures both legal compliance and research reproducibility.

---

[2]https://creativecommons.org/licenses

[3]https://www.gnu.org/licenses/fdl-1.3.en.html

[4]https://opensource.org/license/mit

[5]https://arxiv.org/licenses/nonexclusive-distrib/1.0/license.html

## Acknowledgment

This paper is partially supported by National Natural Science Foundation of China (NO. 62572415, U23A20313, 62372471) and The Science Foundation for Distinguished Young Scholars of Hunan Province (NO. 2023JJ10080).

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

## Contents of Appendix

## A   LLM Usage

Large language models (LLMs) were employed solely for auxiliary purposes, including language polishing, minor code debugging, synthetic data generation, and assistance with experiment evaluation. They were not involved in the research design, the methodological development, or the substantive writing of the article. Consequently, the use of LLMs does not constitute an intellectual contribution to the core content of this work.

## B   GRPO Formalization

Group Relative Policy Optimization (GRPO) (Shao et al., 2024) overcomes a key limitation of traditional reinforcement learning methods, such as PPO (Schulman et al., 2017), which rely on a separate, computationally expensive critic model to estimate value functions. Instead of using a critic, GRPO evaluates policy performance by comparing responses *within groups*, enabling reward-based learning without explicit value estimation and thereby simplifying the training pipeline.

Given an input query $q$, GRPO samples $G$ candidate responses $\{o_1, o_2, \ldots, o_G\}$ from the current (old) policy $\pi_{\theta_{\text{old}}}$. Each response is scored using a predefined reward function, yielding rewards $\{r_1, r_2, \ldots, r_G\}$. To capture *relative* performance within the group, GRPO normalizes these rewards into advantages:

$$A_i = \frac{r_i - \text{mean}(\{r_1, \ldots, r_G\})}{\text{std}(\{r_1, \ldots, r_G\})}, \tag{1}$$

where $\text{mean}(\cdot)$ and $\text{std}(\cdot)$ denote the empirical mean and standard deviation of the group rewards, respectively. During training, GRPO maintains a trainable policy $\pi_\theta$ and a frozen reference model $\pi_{\text{ref}}$. The policy is updated by maximizing the following group-based objective:

$$\mathcal{J}_{\text{GRPO}}(\theta) = \frac{1}{N} \sum_{i=1}^{N} \left( \frac{\pi_\theta(o_i|q)}{\pi_{\theta_{\text{old}}}(o_i|q)} A_i - \beta \cdot \mathcal{KL}\left(\pi_\theta(o_i|q) \,\|\, \pi_{\text{ref}}(o_i|q)\right) \right), \tag{2}$$

where $N$ is the number of responses per group (i.e., $N = G$), and $\beta$ is a hyperparameter controlling the strength of the KL penalty. This objective encourages the policy to assign higher probabilities to responses with above-average rewards within each group, while constraining deviations from the reference model to ensure stable training.

## C   Data

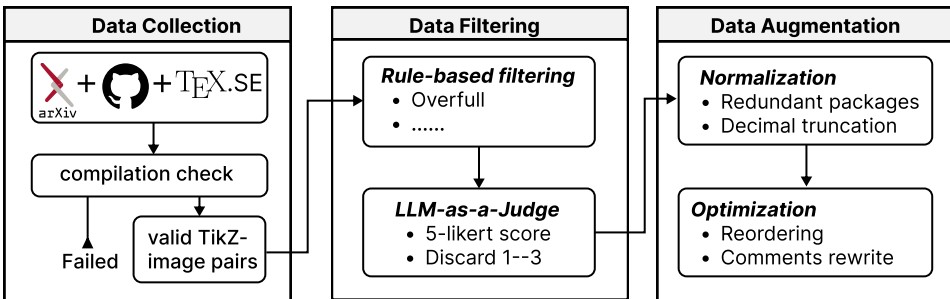

Figure 5: The overall data processing pipeline for constructing the dataset. It consists of three stages: Data Collection, Data Filtering, and Data Augmentation.

## C.1 Data Collection

Our data collection pipeline builds upon the DATi*k*Z series (Belouadi et al., 2024a;b; 2025), which collects Ti*k*Z code from TEX.SE, arXiv papers, and curated GitHub repositories. For arxiv papers, we crawl their source LATEX files and only save those that contain `tikzpicture` environments. We then extract the corresponding Ti*k*Z code blocks via regex matching. Beyond arXiv, our scripts also include routines for collecting data from other sources (TEX.SE, and curated GitHub repositories), ensuring a unified and reproducible pipeline. All extracted snippets are compiled into figures, and samples that fail to compile or render blank images are discarded, yielding an initial dataset of 366,075 valid image–Ti*k*Z code pairs (306,691 from arXiv, 4,961 from curated repositories, and 54,423 from TEX.SE).

## C.2 Data Filtering

Ti*k*Z is a widely adopted drawing language that is not only integrated into LATEX editors but also supported by various graphics platforms and tools. Due to its rich library ecosystem and extensibility, Ti*k*Z can generate figures far beyond regular SVG-like graphics, finding applications in slides, animations, posters, and other contexts. This versatility, however, also leads to a broad spectrum of categories and structural variations in TikZ-generated images. Such diversity inevitably introduces substantial challenges in data quality control.

In this work, we focus exclusively on static, single-page Ti*k*Z figures, and therefore apply a series of filtering steps to discard irrelevant or unsuitable cases. Our data filtering pipeline, illustrated in Fig. 5, proceeds as follows:

1. Rule-based filtering: We exclude low-quality or unsuitable samples according to the following rules:
   (a) Code containing dynamic renderings such as animations, or multi-page layouts outside the target scope;
   (b) Template-based figures with unresolved references (e.g., empty `figure` placeholders) that may mislead models during training;
   (c) Overly cluttered outputs, identified by `pdflatex` logs reporting overfull boxes or out-of-range errors;
   (d) Figures violating density constraints, i.e., more than 16 points per `addplot` series or more than 96 points on a single axis.

   After rule-based filtering, the initial collected dataset is reduced from 366,075 to 258,421 samples.

2. LLM-based filtering: After the rule-based stage, we further use the Qwen2.5-VL-32B model to score each figure on a scale of 1–5, considering information density, structural clarity, stylistic richness, and suitability as training samples. The score distribution demonstrates the high quality of our dataset: 0.07% of samples receive a score of 1, 1.47% score 2, and 11.14% score 3, while the vast majority are rated highly, with 87.14% scoring 4 and 0.17% scoring 5. We filter out all figures with scores 1–3 and retain only those rated 4 or 5, resulting in 225,648 high-quality samples. We also provide the evaluation prompt in Fig. 6.

We now provide an overview of how the initial 366,075 valid TikZ–image pairs were progressively refined into the final optimized TIKZ-30K dataset, accompanied by statistical analysis at each stage. Token statistics are reported separately for pure code tokens (excluding comments) and pure comment tokens. For each stage, we report the minimum and maximum token lengths per sample. Although we do not apply token-length-based filtering before stratified sampling, as even extremely long samples (up to 1,634,716 tokens) can be syntactically valid and compilable, we exclude samples exceeding 14,096 tokens when computing the average, minimum, and maximum token lengths to reduce distortion from outliers.

The results in Table 6 indicate that:

- The stages of *1. Forbidden-keyword filtering*, *4. Decimal-rounding refinement*, *5. LLM-based quality scoring*, and *7. 30K randomly sampled from 58K* show minimal impact on average token counts, indicating that these steps primarily filter or subsample without

Table 6: Statistical evolution across refinement stages. *Note:* #Samp. = number of samples; Code/-Comm. tok. = average pure code/comment tokens; #Long = samples >14,096 tokens (excluded from mean/min/max calc.).

| Stage | #Samp. | Code tok. | Comm. tok. | Min/Max tok. | #Long |
|---|---|---|---|---|---|
| 0. Initial extraction | 366,075 | 731.15 | 11.89 | 25/14,094 | 3,269 |
| 1. Forbidden-keyword filtering | 303,673 | 732.04 | 10.46 | 25/14,094 | 2,947 |
| 2. Density-constraint filtering | 298,699 | 696.10 | 9.57 | 25/14,087 | 1,650 |
| 3. Filtering via `pdflatex` compilation logs | 258,421 | 625.91 | 8.48 | 25/14,087 | 890 |
| 4. Decimal-rounding refinement | 258,421 | 617.29 | 8.48 | 25/14,087 | 888 |
| 5. LLM-based quality scoring | 225,648 | 617.97 | 8.47 | 25/14,087 | 746 |
| 6. Unequally spaced token-bin samp. (58K) | 58,000 | 758.44 | 11.58 | 32/4,092 | — |
| 7. Random samp. (30K) | 30,000 | 758.71 | 11.40 | 35/4,091 | — |
| 8. Final Ti*k*Z30K | 29,859 | 777.85 | 84.56 | 60/4,178 | — |

substantially altering the underlying code complexity distribution. This aligns with our expectations.

- *2. Density-constraint filtering* significantly reduces the average code token length (from ∼732 to ∼696), while *3. Filtering via* `pdflatex` *compilation logs* causes a further substantial decrease (to ∼626 tokens). **These notable changes are compliant with our expectations**, as the removed samples often represent data points that exceed the reasonable scope of the diagram-to-code task. For instance, the density filter removes diagrams with overly dense coordinate data (e.g., attempting to accurately reconstruct positions for over 100 points). Similarly, the `pdflatex` log filtering eliminates cases leading to truncated images due to LaTeX overfull errors, where content extends beyond the defined page margins and is consequently clipped during compilation. Removing these overly complex or technically flawed samples ensures the dataset's focus on learnable and reproducible diagram-to-code mappings.

- *6. Unequally spaced token-bin sampling* intentionally reshapes the token-length distribution, which explains the increase in average pure code tokens from ∼618 to 758.

- *8. Final optimized TikZ30K* results in a dramatic increase in comment token length (from ∼11.4 to 84.6 on average), as intended.

Overall, no unexpected data distortion was introduced throughout the pipeline; all observed changes in token statistics are consistent with our design intentions, reflecting targeted filtering of overly complex or invalid samples beyond the scope of the diagram-to-code task.

**Prompt for Figure Quality Score Judgment**

**System Prompt:**
You are an expert in image quality evaluation and LLM training data curation. Your task is to rate the visual quality of technical figures (rendered from TikZ code). Your evaluation will be used to construct a high-quality dataset for improving multimodal LLMs' performance on Image-to-TikZ code generation. Focus only on the image itself. Do not assume the figure must follow a specific type (chart, math diagram, flowchart, etc.).

**User Prompt:**
Here is a figure. Please analyze it and give a score from 1 to 5 according to the following rules:

**1 (Very Poor — Not usable for training):**
The figure is visually incoherent or severely degraded. Key elements (text, shapes, lines, labels) are unreadable, missing, or so cluttered that no meaningful structure can be perceived. May appear as random fragments, noise, or broken rendering. No reliable visual cues for an MLLM to learn from.

**2 (Poor — Marginal training value):**
The figure is structurally incomplete, visually confusing, or overly simplistic. — If overly simple (e.g., a single shape, minimal text, trivial layout), it provides no meaningful structural or stylistic patterns for an MLLM to learn from, even if rendered correctly. — If visually degraded (e.g., overlapping elements, unreadable text, broken layout), it lacks sufficient reliable cues for the model to reconstruct the underlying TikZ structure. In either case, the image fails to serve as a useful training sample: it is either too trivial to be instructive or too corrupted to be recoverable.

**3 (Fair — Acceptable but needs improvement):**
The figure conveys a clear structure and contains readable elements, but suffers from noticeable visual flaws: — Text is cramped or low-contrast; — Layout is crowded or unbalanced; — Some components are ambiguous or poorly spaced. Suitable for training only with caution — visual cues exist, but are suboptimal.

**4 (Good — Strong training sample)**
The figure is visually clear, well-organized, and complete. Minor imperfections may exist (e.g., slightly tight spacing, non-optimal label placement), but all key elements are legible and logically arranged. Provides sufficient, unambiguous visual information for an MLLM to learn structural and stylistic patterns from.

**5 (Excellent — Ideal training sample)**
The figure is visually polished, highly legible, and structurally exemplary. All elements (text, lines, shapes, labels) are crisp, well-spaced, and harmoniously arranged. Color, contrast, and layout support immediate comprehension. Represents the gold standard for what an Image-to-TikZ model should aim to reconstruct.

**Output Format (always follow this structure):**
**Description:** 1–2 sentences of what the figure shows.
**Overall evaluation:** Write 2 to 4 sentences explaining the overall quality and why you assigned this score.
**Score:** Final rating in the format **Score: X**

Figure 6: Prompt template for figure quality score judgment.

## C.3 Data Category Distribution

To better characterize the dataset after filtering, we employ the Qwen2.5-VL-32B model to perform large-scale categorization of all rendered figures. The resulting distribution, presented in Fig. 7, demonstrates that our collection spans a broad spectrum of TikZ figure types, including flowcharts, scientific diagrams, plots, and graph structures. This categorical diversity ensures that the dataset not only captures a wide variety of visual and structural patterns but also provides richer supervision signals. Consequently, such heterogeneity is expected to enhance the robustness and generalization capacity of the downstream model. We provide image examples of different categories in the collected TikZ dataset in Fig. 8 and the complete classification prompt in Fig. 9.

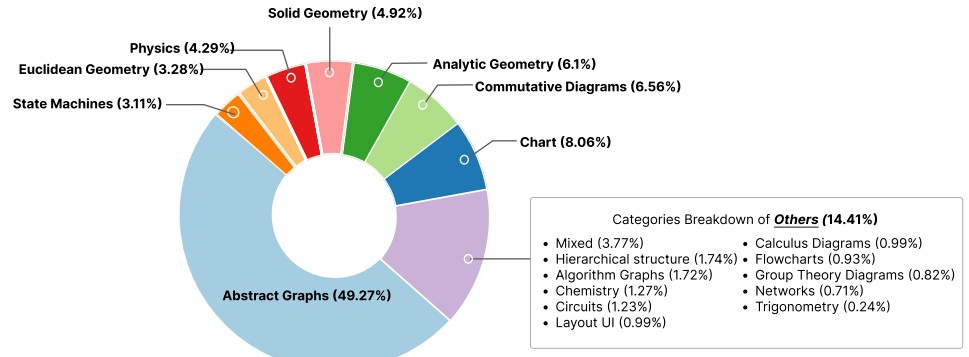

Figure 7: Overall category distribution of the collected Ti*k*Z dataset.

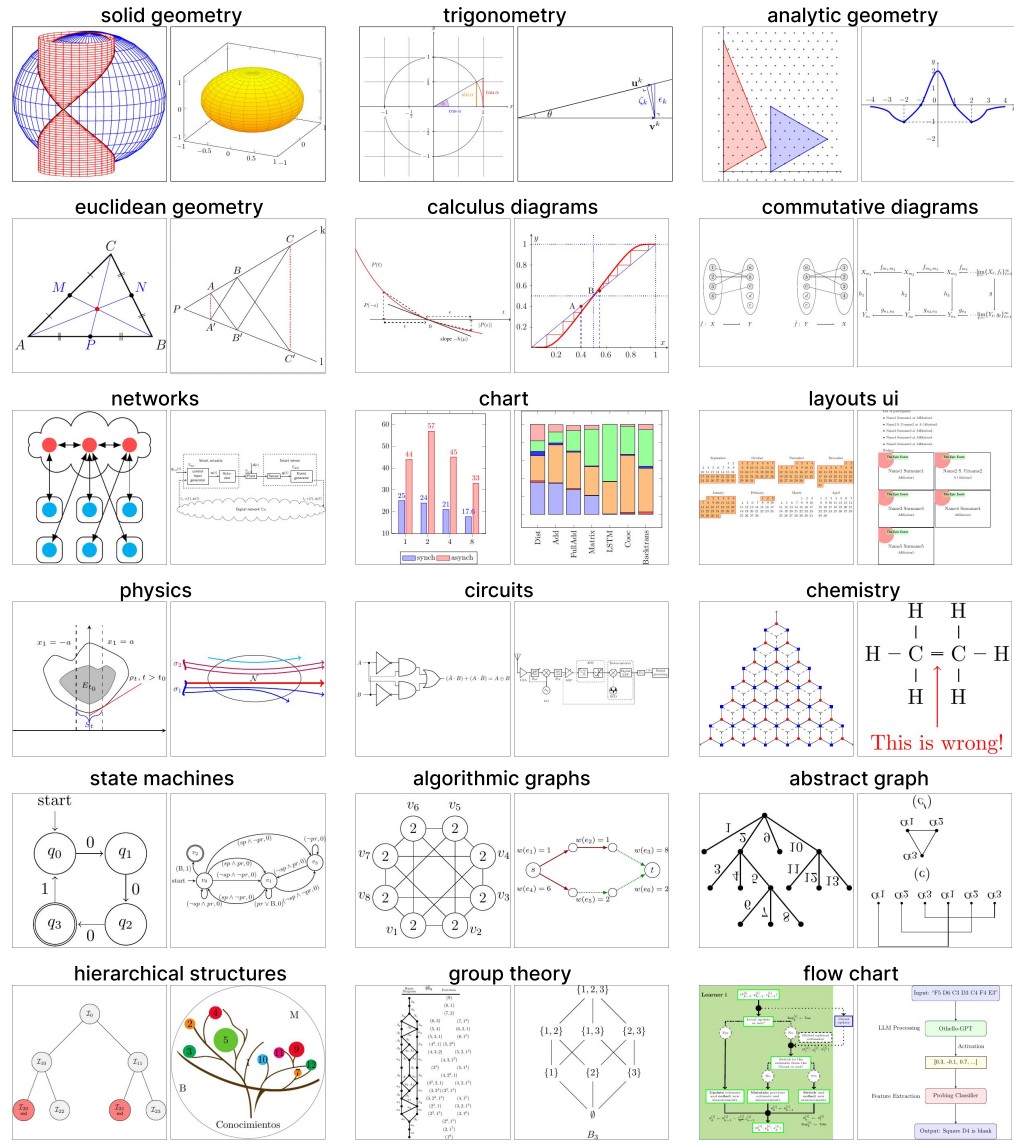

Figure 8: Examples of different categories in the collected Ti*k*Z dataset.

## Prompt for Category Classification

**System Prompt:**
You are an expert classifier for technical figures. Given ONE image, your task is to assign it to exactly ONE of the most specific categories from the predefined taxonomy. Focus only on the image itself without making assumptions about its context.

**User Prompt:**
Here is a figure. Please determine the most appropriate category according to the following taxonomy:

**Categories with guidance:**
- analytic geometry: coordinate systems, function plots, vectors, points
- calculus diagrams: limits, derivatives, tangents, integrals, Riemann sums
- euclidean geometry: constructions with shapes, angles, proofs
- solid geometry: 3D solids, polyhedra, cones, spheres
- trigonometry: unit circle, triangles, trigonometric functions
- abstract graphs: nodes and edges, connectivity, theoretical graphs
- algorithmic graphs: shortest paths, spanning trees, flows
- state machines: finite automata, states, labeled transitions
- commutative diagrams: category theory objects and arrows
- group theory diagrams: Cayley graphs, subgroup lattices
- flowcharts: process steps, decision diamonds, arrows, sequences
- hierarchical structures: trees, org charts, syntax trees
- networks: computer or data network topologies
- chemistry: chemical structures, reactions, mechanisms
- circuits: electronic schematics, logic gates
- layouts ui: UI wireframes, document layouts, dashboards
- physics: free-body diagrams, optics rays, electromagnetism, Feynman/tensors
- chart: scientific charts with axes, ticks, legends
- mixed: ambiguous or multi-type diagrams combining categories

**Decision Rules:**
Choose the single most specific category; prefer domain-specific (e.g., physics, chemistry) over generic (e.g., graphs); use `mixed` if multiple distinct types coexist.

**Output Format:**
`{"label":  "<category>", "brief":  "<=20 words"}`

Figure 9: Prompt template for category classification.

## C.4 Code Augmentation

We tokenize all Ti*k*Z code snippets from the filtered dataset using the Qwen-2.5-VL series tokenizer and analyze their token-length distribution. As shown in Figure 10, the majority of figures contain fewer than 600 tokens. To ensure balanced representation across different complexity levels, we apply stratified sampling based on predefined token-count bins: (0, 200], (200, 400], (400, 600], (600, 800], (800, 2000], and (2000, 4096]. This process yields a total of 58,000 training samples. These are then randomly divided into 30,000 samples for SFT and 28,000 samples for RL.

**Normalization.** We perform a manual audit of the collected Ti*k*Z code snippets, focusing on drawing order, organizational structure, and the presence of redundant or unnecessary package or style declarations. Our analysis reveals several recurring issues: (1) disorganized code structure lacking clear modularity, with node declarations and drawing commands arranged in ways inconsistent with human cognitive expectations; (2) redundant package imports and style definitions; and (3) insufficient or absent annotations, which obscure the intended purpose and functionality of the code; and (4) excessively long decimal representations. To normalize the decimal representations, we round each decimal number to four decimal places. For example, `10.000000001` is rounded to `10.00`.

**Optimization.** To further enable our model to internalize code organization and drawing sequences that mirror human reasoning habits, we leverage the Qwen3-Coder-480B-A35B-Instruct model to systematically reorder and refine code samples. During the process, redundant references are removed to reduce noise, and scaffold-like comments are inserted in a natural, incremental style. These comments act as both inline explanations and cognitive signposts of human drawing intent. This combination of structural refinement and reasoning-oriented commentary yields clearer exemplars for training and helps bridge the gap between raw Ti*k*Z code and the natural planning patterns humans follow when constructing figures.

**Post-Verification.** After optimization, the refined code is re-rendered to generate a new image. We then compare this rendered output against the original figure using the mean squared error (MSE) to quantify similarity. For cases where rendering fails or where the visual discrepancy exceeds a predefined threshold, the code is passed back to the model for iterative correction. This loop continues until the code compiles successfully and the rendered image aligns with, or closely approximates, the original, with a maximum of three iterations. For example, after three rounds of iterative refinement on 30000 samples, the overall success rate improved from 94.45% to 98.28%, and finally to 99.53%, yielding a final high-quality dataset of 29,859 samples. This post-verification stage ensures consistency between optimized code and the intended visual semantics of the original figure.

As discussed in our main results, the incorporation of reordered and annotated code samples, which serve as implicit "planning scaffolds", leads to substantial improvements in model performance. Examples of code augmentation are shown in Fig. 11, Fig. 12, and Fig. 13 to illustrate our approach. The exact prompting strategy is provided in Fig. 14.

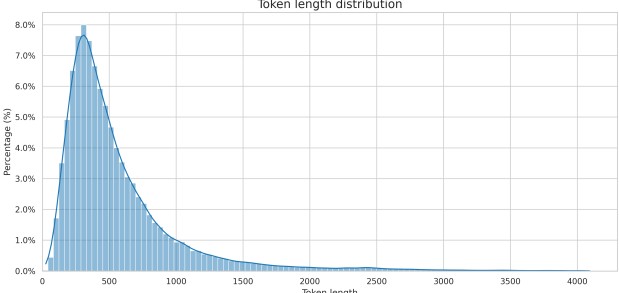

Figure 10: Token length distribution of the dataset after filtering. The distribution is strongly dominated by short sequences (fewer than 600 tokens), leading to a skewed and imbalanced dataset. Such an imbalance may bias the model toward short inputs that typically represent simpler visual patterns.

### C.4.1 Example 1

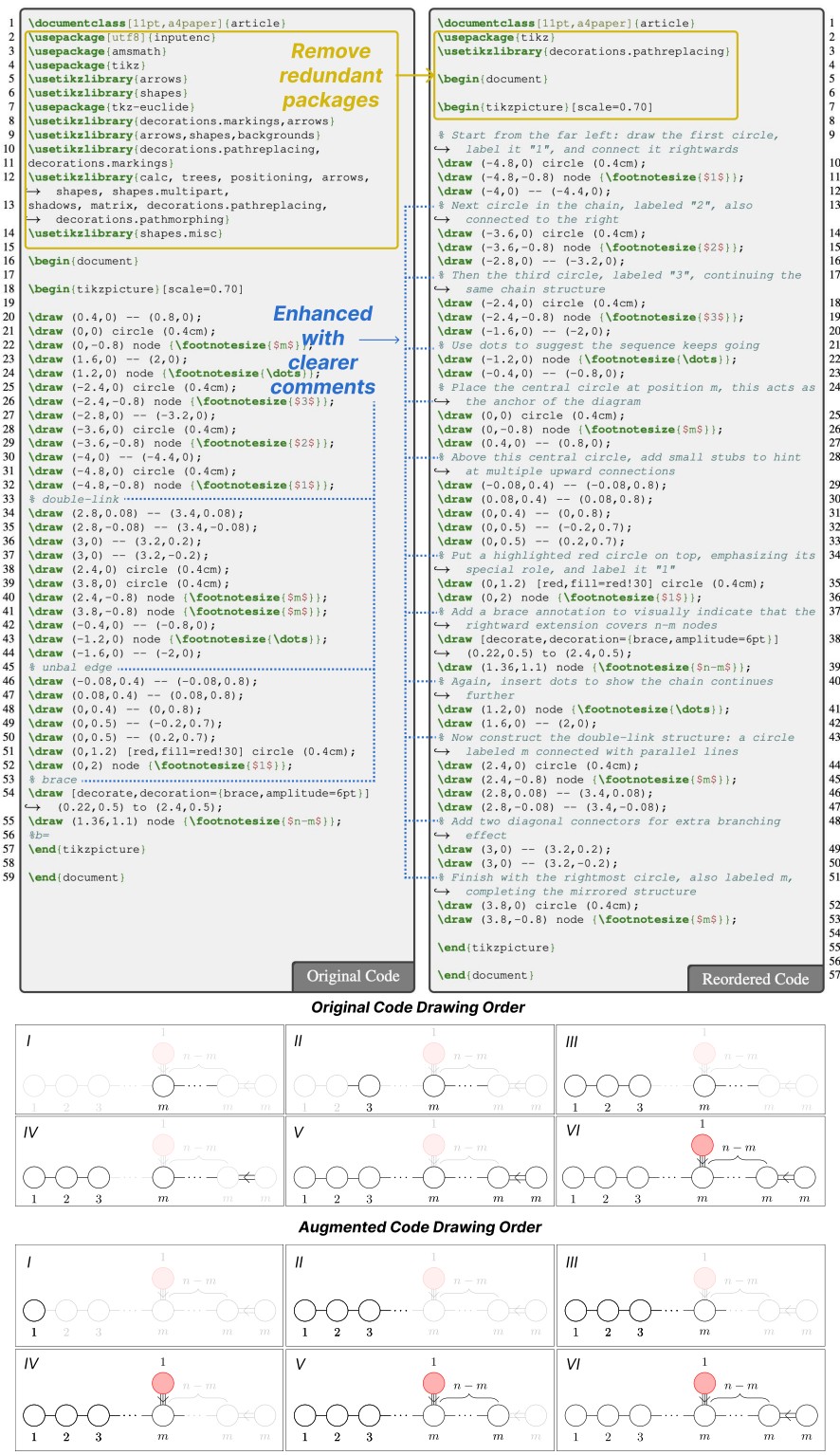

Figure 11: Comparison before and after data augmentation. Block-by-block rendering of the sequential diagram after reordering, showing the resolved ordering issue.

## C.4.2 Example 2

```latex
\documentclass[a4paper,12pt]{article}
\usepackage{cite, amsmath, amssymb}
\usepackage{tikz}

\begin{document}

\begin{tikzpicture}

  [scale=0.8,auto=left,every
↪   node/.style={circle,scale=0.7}]

  \node[draw,circle,...] (o) at (2.3,4) {};
  \node[draw,circle,...] (n) at (1.5,4) {};
   \node[draw,circle,...] (q) at (0.75,4) {};
  \node[draw,circle,...] (v) at (3,4) {};
  \node[draw,circle,...] (v2) at (3.5,4) {};
  \node[draw,circle,...] (z) at (4.2,4) {};
   \node[draw,circle,...] (z2) at (4.8,4) {};
     \node[draw,circle,...] (y3) at (5.4,4) {};
        \node[draw,circle,...] (y4) at (6,4) {};
  \node[draw, circle,...] (m) at (1.7,5) ...;
  \node[draw,circle,...] (l) at (3,5) {$\otimes$};
  \node[draw,circle,...] (y) at (4.6,5) ...;
   \node[draw,circle,...] (y2) at (5.4,5) ...;
  \node[draw,circle,...] (k) at (4.6,6) {$\cup$};
  \node[draw, circle, ...] (j) at (2,6) {$\cup$};
  \node[draw,circle,...] (h) at (2,7) {$\otimes$};
  \node[draw, circle, ...] (g) at (0,6) {$\cup$};
  \node[draw,circle,...] (f) at  (0.6,5) ...;
  \node[draw,circle,...] (a) at (-1,5) {$\otimes$};
  \node[draw, circle, ...] (b) at (-1.4,4) {};
  \node[draw,circle,...] (c) at (-0.75,4) {};
  \node[draw,circle,...] (e) at (0,4) {};

  \path
  (a) edge node[left]{} (b)
        (a) edge node[below]{} (c)

   (f) edge node[below]{} (e)
      (k) edge node[below]{} (y)
       (k) edge node[below]{} (y2)
        (y) edge node[below]{} (z)
         (y) edge node[below]{} (z2)
            (y2) edge node[below]{} (y3)
              (y2) edge node[below]{} (y4)
                 (l) edge node[below]{} (v)
     (l) edge node[below]{} (v2)
       (f) edge node[right]{}(g)
        (g) edge node[left]{}(a)
       (f) edge node[left]{}(q)
       (h) edge node[right]{}(j)
       (h) edge node[left]{}(g)
       (h) edge node[left]{}(k)
       (j) edge node[right]{}(l)
       (j) edge node[below]{}(m)
       (m) edge node[below]{}(n)
       (m) edge node[right]{}(o)
       ;
\end{tikzpicture}

\end{document}
```

**Original Code**

```latex
\documentclass[a4paper,12pt]{article}
\usepackage{tikz}

\begin{document}

\begin{tikzpicture}[scale=0.8,auto=left,every
↪   node/.style={circle,scale=0.7}]

  % Start with the bottom row: place a sequence of
↪   black circles as the foundation
  \node[draw,circle,...] (b) at (-1.4,4) {};
  \node[draw,circle,...] (c) at (-0.75,4) {};
  \node[draw,circle,...] (e) at (0,4) {};
  ...

  % Add a middle layer above: light blue circles,
↪   some carrying $\otimes$ symbols
  \node[draw,circle,...] (a) at (-1,5) {$\otimes$};
  \node[draw,circle,...] (f) at (0.6,5) {$\otimes$};
  \node[draw,circle,...] (m) at (1.7,5) {$\otimes$};
  ...

  % Place the top row: connectors with $\cup$ and
↪   $\otimes$ to emphasize branching roles
  \node[draw,circle,...] (g) at (0,6) {$\cup$};
  \node[draw,circle,...] (j) at (2,6) {$\cup$};
  \node[draw,circle,...] (k) at (4.6,6) {$\cup$};
  ...

  % Connect nodes to reveal the intended structure
  \path
    % a links downward to b and sideways to c
    (a) edge (b)
    (a) edge (c)
    % f connects to e, g, and also to q
    (f) edge (e)
    (f) edge (g)
    (f) edge (q)
    % g loops back into a
    (g) edge (a)
    % k branches to y and y2, which extend further
↪     to the right
    (k) edge (y)
    (k) edge (y2)
    (y) edge (z)
    ...
    % l connects downward to v and v2
    (l) edge (v)
    (l) edge (v2)
    % h ties together j, g, and k as a hub
    (h) edge (j)
    (h) edge (g)
    (h) edge (k)
    % j bridges toward l and also drops into m
    (j) edge (l)
    (j) edge (m)
    % m anchors back down into the baseline at n and
↪     o
    (m) edge (n)
    (m) edge (o)
  ;
\end{tikzpicture}

\end{document}
```

**Reordered Code**

Figure 12: Comparison before (left) and after (right) reordering.

23

### C.4.3 Example 3

Original Code (left):

```latex
\documentclass{amsart}
\usepackage{amssymb,enumerate,bbm,amsmath}
\usepackage[colorlinks=true,linkcolor=blue,
-> citecolor=blue]{hyper ref}
\usepackage{tikz}

\begin{document}

\begin{tikzpicture}[x=0.75pt,...]

        \draw [color=...]   (195.99,203.13) --
        ↪ (392.3,202.2) ;
        \draw [color=... ]   ... (375.05,202.19) ;
        \draw [color=... ]   ... (10.93,3.29)   ;
        \draw [color=... ]   ... (10.93,3.29)   ;
        \draw [color=... ]   (237.78,202.74) --
        ↪ (237.98,225.26) ;
        \draw [shift=...] [color=... ][line
        ↪ width=0.75]   ... (10.93,3.29)   ;
        \draw [color=... ]   ...-- (216.05,202.83) ;
        \draw [shift=...] [color=... ][line
        ↪ width=0.75]   ... (10.93,3.29)   ;
        \draw [color=...]   ...-- (219.7,186.82) ;
        \draw [shift=...] [color=... ][line
        ↪ width=0.75]   ... (10.93,3.29)   ;
        \draw  [color=... ][fill=... ] ... -- cycle
        ↪ ;
        \draw  [color=...][fill=... ] ... -- cycle ;
        \draw  [color=... ][fill=... ] ... -- cycle
        ↪ ;
        \draw  [color=... ] ...  -- cycle ;

        % Text Node
        \draw (218.51,170.47) node ... ;
        % Text Node
        \draw (202.72,185.79) node ... ;
        % Text Node
        \draw (228.46,227.76) node ... ;
        % Text Node
        \draw (206.8,211.45) node ... ;
        % Text Node
        \draw (333.24,144.43) node ... ;
        % Text Node
        \draw (270.17,153.19) node ...;
        % Text Node
        \draw (319.64,206.02) node ... ;
        % Text Node
        \draw (284.81,204.98) node ... ;

\end{tikzpicture}

\end{document}
```

Reordered Code (right):

```latex
\documentclass{amsart}
\usepackage{amssymb,amsmath}
\usepackage{tikz}

\begin{document}

\begin{tikzpicture}[x=0.75pt,y=0.75pt,yscale=-1
,xscale=1]

% First draw the background ellipse as the frame
\draw  [color=... ] ... -- cycle ;

% Then place the main red horizontal line across the
↪ ellipse
\draw [... ]...-- (392.3,202.2) ;

% Again add the green curved path above the baseline
\draw [color=...] ... (375.05,202.19) ;

% Now draw arrows radiating out from the left anchor
↪ point
\draw [color=...] ... -- (221.14,220.13) ;
\draw [shift=...][color=... ]... (10.93,3.29) ;

\draw [color=...]...-- (237.98,225.26) ;
\draw [shift=...][color=... ]... (10.93,3.29) ;

\draw [color=...] ...(216.05,202.83) ;
\draw [shift=...][color=... ]... (10.93,3.29) ;

\draw [color=...] ...-- (219.7,186.82) ;
\draw [shift=...][color=...]... (10.93,3.29) ;

% Then add a vertical arrow on the right for balance
\draw [color=...] ...-- (367.49,157.22) ;
\draw [shift=...][color=...] ... (10.93,3.29) ;

% Next highlight some key points with filled circles
\draw [color=...] ...-- cycle ;
\draw [color=...] ...-- cycle ;
\draw [color=...] ... -- cycle ;

% Finally add text labels near the arrows and points
\draw (218.51,170.47) node ...;
\draw (202.72,185.79) node ...;
\draw (228.46,227.76) node ...;
        ...

\end{tikzpicture}

\end{document}
```

**Original Code Drawing Order**

**Augmented Code Drawing Order**

Figure 13: Comparison before (left) and after (right) reordering. For clarity, we omit parts of the original code and focus only on the drawing order.

## Prompt for Code Augmentation

**System Prompt:**
You are an expert in TikZ code refinement and ordering. Your task is to adjust the TikZ code to follow a natural and intuitive plotting sequence while preserving the rendered image content.

**User Prompt:**
Here is a TikZ code snippet. Please refine it according to the following rules:

• Code order:
Change the TikZ code order if the plotting sequence is not natural or intuitive. Do not impact the rendered image content when altering the code.

• Variables and macros:
Pay attention to variable and macro declarations. No variable or macro may be referenced before it is defined. If reordering changes a reference, carefully transform the position expressions. You need to understand the coordinate rules of TikZ.

• Coordinate rules:
When you define an object at $(x, y)$, the coordinate refers to the *center* of the object. However, when you use relative positions such as 'left' or 'right' of an object, it refers to the *edge* of the object.

• Styles:
If the code defines styles using `tikzset` or `tikzstyle`, check if they are used. Remove unused styles, just as you would remove unused imports.

**Output Format:**
• Thinking: Briefly explain your reasoning (1–2 sentences).
• Code: Return the refined TikZ code wrapped in triple backticks.

Figure 14: Prompt template for code augmentation.

# D Evaluation Details

## D.1 Evaluation Prompt

Here we present the evaluation prompt for the Image-to-TikZ generation task.

---

**Prompt for Evaluating Image-to-TikZ Generation (Belouadi et al., 2024b)**

This is a picture of a scientific figure. Generate LaTeX code that draws this scientific figure using TikZ. Ensure that the LaTeX code is self-contained and does not require any packages except TikZ-related imports. Don't forget to include \usepackage{tikz}! In your answer, use a ```latex code block to wrap the code part.

---

## D.2 Results Showcase

Here we present a subset of the input–output pairs from our Image-to-TikZ generation task. These examples illustrate the effectiveness of our approach in converting scientific diagrams into self-contained LaTeX code with TikZ.

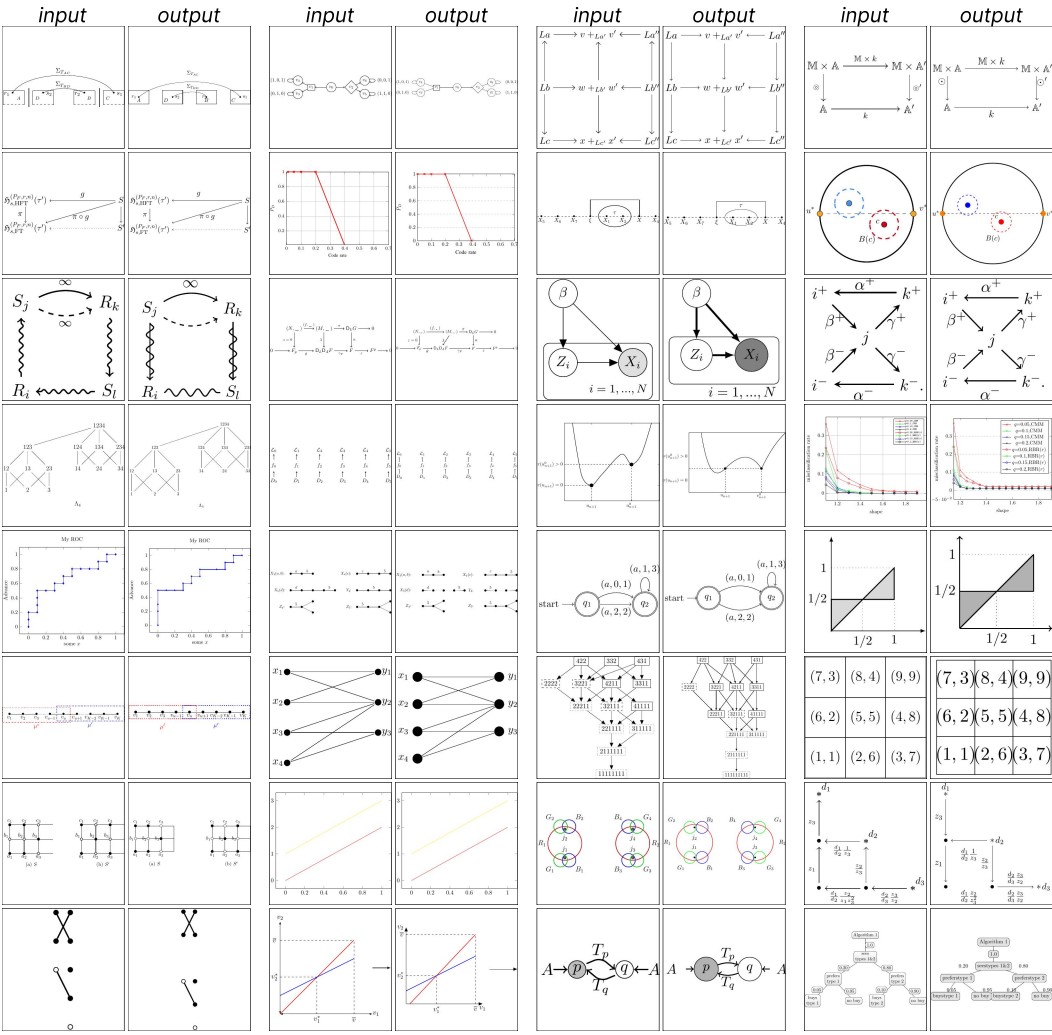

Figure 15: Showcase of Image-to-TikZ generation results. Each pair shows the input figure (left) and the corresponding generated TikZ output (right).

### D.3 Code Efficiency Metric

Here, we report two code efficiency metrics in Table 7, supplementing the Table 1 in the main content. We present the 10% winsorized mean token and character length, following the previous study (Belouadi et al., 2024b) to maintain consistency. Winsorization replaces the top and bottom 10% of values with the 10th and 90th percentiles, reducing outlier influence while keeping the sample size.

We observe that the most capable models in Image-to-TikZ generation, Gemini-2.5-Pro-Thinking (1640 chars, 914 tokens) and DaVinci-7B (1618 chars, 983 tokens), exhibit similar code efficiency. Notably, the other three proprietary MLLMs all produce shorter outputs, indicating higher code efficiency, with GPT-5-Default (1081 chars, 604 tokens) achieving the most concise results. Among specialized TikZ-generation MLLMs, DaVinci-7B achieves the highest code efficiency and is similar to its base model, Qwen2.5-VL-7B. Moreover, code efficiency improves with model scale, as demonstrated by the Qwen2.5-VL series (7B, 32B,72B). This trend suggests that scaling up model size may enhance code efficiency.

Table 7: Compile results of DaVinci against different models.

| Model | Average character length | Average token count |
|---|---|---|
| *Proprietary MLLMs* | | |
| Gemini2.5-Pro-Thinking | 1640.07 | 914.24 |
| GPT-5-Default | **1081.45** | **603.93** |
| Claude-Sonnet-4 | 1320.93 | 778.69 |
| Claude-Sonnet-4-thinking | 1243.53 | 727.16 |
| *Open-Sourced MLLMs* | | |
| Qwen2.5-VL-7B | 1687.20 | 1132.25 |
| Qwen2.5-VL-32B | 1186.80 | 653.85 |
| Qwen2.5-VL-72B | 1107.85 | 615.94 |
| *Specialized TikZ-generation MLLMs* | | |
| DetikZify-V2-8B | 2357.65 | 1337.33 |
| DiagramAgent-7B | 1955.69 | 1191.66 |
| DaVinci-SFT-7B | 1722.87 | 984.38 |
| DaVinci-7B | 1618.04 | 983.01 |

## D.4 Additional Human Evaluation Details

### D.4.1 Interface for Human Evaluation

We developed a simple online annotation interface for conducting human evaluation, as shown in Figure 16. We invited six graduate students as external annotators (2 male and 4 female), each of whom is tasked to finish two groups of annotations.

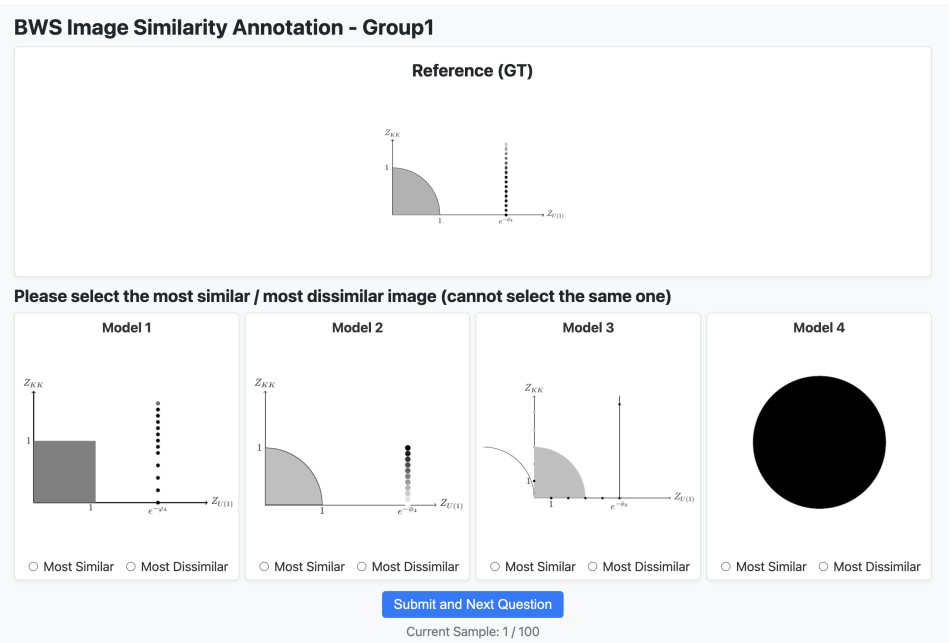

Figure 16: The interface used for human evaluation.

### D.4.2 Agreement Analysis

We measure agreement among annotators using split-half reliability (SHR). For each study, we randomly split annotators into two groups. For each image, we aggregate the Best-Worst Scaling annotations within each group to obtain vectors of model scores, compute the Spearman rank correlation $\rho$ between the groups, and report the mean correlation across all images. The resulting SHR values of $\rho = 0.7227$ for Group 1 and $\rho = 0.7878$ for Group 2 indicate strong inter-annotator agreement, supporting the reliability of our human evaluation results.

We further evaluate the agreement between image similarity metrics and human judgments. Specifically, we derive rankings from image similarity metric scores and compute Spearman rank correlations against human-derived rankings. Following Belouadi et al. (2024b), we report these correlation values at both the instance level and the group level. At the group level, we find that the employed metrics exhibit strong alignment with aggregated human judgments, indicating their effectiveness in evaluating overall model performance.

Table 8: Correlations of image similarity metrics with humans at the instance and group level.

| Metric | Instance-level | | | Group-level | | |
|---|---|---|---|---|---|---|
| | Group 1 | Group 2 | Mean | Group 1 | Group 2 | Mean |
| LPIPS | 0.368 | 0.333 | 0.350 | 0.949 | 1.000 | 0.974 |
| DSIM | 0.350 | 0.507 | 0.428 | 0.949 | 1.000 | 0.974 |
| SigLIP | 0.352 | 0.520 | 0.436 | 0.949 | 1.000 | 0.974 |
| MSE | 0.358 | 0.209 | 0.284 | 0.949 | 0.800 | 0.874 |
| SSIM | 0.152 | 0.438 | 0.295 | 0.633 | 0.800 | 0.716 |

## D.5 Additional Automatic Evaluation

We construct an additional test set comprising 2,000 randomly sampled samples from newly collected data sourced from arXiv, restricted to papers published after January 2025, and licensed for redistribution. This part of the data source is not considered in our previous data construction process. The results are presented in Table 9. The overall performance is consistent with findings on the official test set, though this new set appears to be somewhat easier, likely due to its random sampling process, in contrast to the curated, human-selected nature of the DAT$\iota k$Z$_{v3}$ test set.

Table 9: Evaluation results of 2000 additional test samples. **Bold** values denote the best scores. Arrows $\uparrow\downarrow$ indicate metric directionality.

| Model | Code | | | Image | | | | |
|---|---|---|---|---|---|---|---|---|
| | Pass@1$\uparrow$ | TED$\downarrow$ | cBLEU$\uparrow$ | DSIM$\uparrow$ | SigLIP$\uparrow$ | SSIM$\uparrow$ | MSE$\downarrow$ | LPIPS$\downarrow$ |
| *Open Sourced MLLMs* | | | | | | | | |
| Qwen2.5-VL-7B | 78.20 | 54.45 | 6.52 | 77.05 | 90.30 | 74.44 | 82.62 | 27.40 |
| Qwen2.5-VL-32B | 83.20 | 52.97 | 7.62 | 77.92 | 90.66 | 75.06 | 80.02 | 26.65 |
| Qwen2.5-VL-72B | 86.30 | 52.71 | 7.66 | 80.52 | 92.12 | 74.52 | 78.09 | 26.17 |
| GLM-4.5V-106B-A12B | 76.75 | 53.51 | 6.25 | 80.78 | 91.95 | 75.45 | 77.72 | 24.97 |
| GLM-4.5V-Thinking | 67.10 | 54.92 | 3.59 | 81.15 | 92.43 | **77.13** | 77.58 | 23.72 |
| *Specialized TikZ-generation MLLMs* | | | | | | | | |
| DetikZify-V2-8B | 86.30 | 51.46 | 10.28 | 85.78 | 93.13 | 76.95 | 67.32 | 22.00 |
| DiagramAgent-7B | 71.85 | 53.45 | 8.57 | 73.60 | 88.51 | 73.97 | 84.59 | 28.39 |
| DaVinci-SFT-7B | 89.80 | **50.48** | **11.25** | 85.32 | 93.87 | 76.24 | 69.77 | 22.86 |
| DaVinci-7B | **99.36** | 52.33 | 9.42 | **86.88** | **94.50** | 75.90 | **61.44** | **20.80** |

## D.6 Additional Data Ablation Study

To further validate the effectiveness of our data curation strategy, we conduct a controlled ablation study on the public version of DAT$\iota$kZ$_{v3}$[6], which contains 145,366 samples. Applying our filtering and optimization pipeline, we derive a refined subset of 112,177 samples, denoted as DAT$\iota$kZ$_{v3}$-optimized. We perform supervised fine-tuning (SFT) on both datasets using the same configuration as DaVinci-7B-SFT, with Qwen2.5-VL-7B as the base model.

The evaluation results on the test set of DAT$\iota$kZ$_{v3}$-public are presented in Table 10, demonstrating that our curation strategy significantly improves Pass@1, increasing from 67.7% to 84.9%, while maintaining or slightly enhancing other key metrics, with a 22.83% reduction in dataset size.

Table 10: Results of the additional ablation study on the impact of our data curation strategy.

| Model | Pass@1$\uparrow$ | cBLEU$\uparrow$ | DSIM$\uparrow$ | SigLIP$\uparrow$ | SSIM$\uparrow$ | MSE$\downarrow$ |
|---|---|---|---|---|---|---|
| Qwen2.5-VL-7B + DAT$\iota k$Z$_{v3}$-public | 67.7% | 3.98 | 82.56 | 93.17 | **73.85** | **71.13** |
| Qwen2.5-VL-7B + DAT$\iota k$Z$_{v3}$-optimized | **84.9%** | **7.19** | **82.95** | **93.90** | 72.75 | 72.01 |

---

[6]`https://huggingface.co/datasets/nllg/datikz-v3`

# E  Reward Design

## E.1  Spatio-Textual Similarity Rewarding

Algorithm 1 describes the complete calculation process of spatio-textual similarity rewarding.

---

**Algorithm 1** Spatio-Textual Similarity Rewarding

---

 1: **Input:** Predicted boxes $B_{pred}$, predicted texts $T_{pred}$.
 2: **Input:** Ground-truth boxes $B_{gt}$, ground-truth texts $T_{gt}$.
 3: **Output:** Scalar reward score $R$.
 4: Initialize set of matches $M \leftarrow \emptyset$.
 5: Initialize remaining indices $P_{rem} \leftarrow \{1, \ldots, |B_{pred}|\}$, $G_{rem} \leftarrow \{1, \ldots, |B_{gt}|\}$.
   ▷ *Stage 1: Exact Text Matching*
 6: **for** $j \in G_{rem}$ **do**
 7:     Find candidates $P_{cand} \leftarrow \{i \in P_{rem} \mid \mathrm{lower}(T_{pred}[i]) = \mathrm{lower}(T_{gt}[j])\}$.
 8:     **if** $|P_{cand}| > 0$ **then**
 9:         Find best match $p^* \leftarrow \arg\max_{i \in P_{cand}} d_{IoU}(B_{pred}[i], B_{gt}[j])$.
10:         $M \leftarrow M \cup \{(p^*, j)\}$.
11:         $P_{rem} \leftarrow P_{rem} \setminus \{p^*\}$, $G_{rem} \leftarrow G_{rem} \setminus \{j\}$.
12:     **end if**
13: **end for**
   ▷ *Stage 2: Approximate Text Matching*
14: **for** $j \in G_{rem}$ **do**
15:     Find candidates $P_{cand} \leftarrow \emptyset$.
16:     **for** $i \in P_{rem}$ **do**
17:         $d_L \leftarrow \mathrm{Levenshtein}(T_{pred}[i], T_{gt}[j])$.
18:         $\tau \leftarrow \mathrm{AdaptiveThreshold}(T_{pred}[i], T_{gt}[j])$.
19:         **if** $d_L \leq \tau$ **then**
20:             Add $(i, d_L)$ to $P_{cand}$.
21:         **end if**
22:     **end for**
23:     **if** $|P_{cand}| > 0$ **then**
24:         $d_{min} \leftarrow \min_{(i, d_L) \in P_{cand}} d_L$.
25:         $P_{best\_cand} \leftarrow \{i \mid (i, d_{min}) \in P_{cand}\}$.
26:         $p^* \leftarrow \arg\max_{i \in P_{best\_cand}} d_{IoU}(B_{pred}[i], B_{gt}[j])$.
27:         $M \leftarrow M \cup \{(p^*, j)\}$.
28:         $P_{rem} \leftarrow P_{rem} \setminus \{p^*\}$, $G_{rem} \leftarrow G_{rem} \setminus \{j\}$.
29:     **end if**
30: **end for**
   ▷ *Final Reward Calculation*
31: $total\_score \leftarrow 0$.
32: **for** each matched pair $(p, g)$ in $M$ **do**
33:     $score \leftarrow (d_{IoU}(B_{pred}[p], B_{gt}[g]) + 1)/2$.
34:     $total\_score \leftarrow total\_score + score$.
35: **end for**
36: $R \leftarrow total\_score / \max(|B_{pred}|, |B_{gt}|)$.
37: **return** $R$.

---

Figure 17 presents visualizations of the Spatio-Textual match results between the ground truth and model output during training, spanning cases from simple to complex.

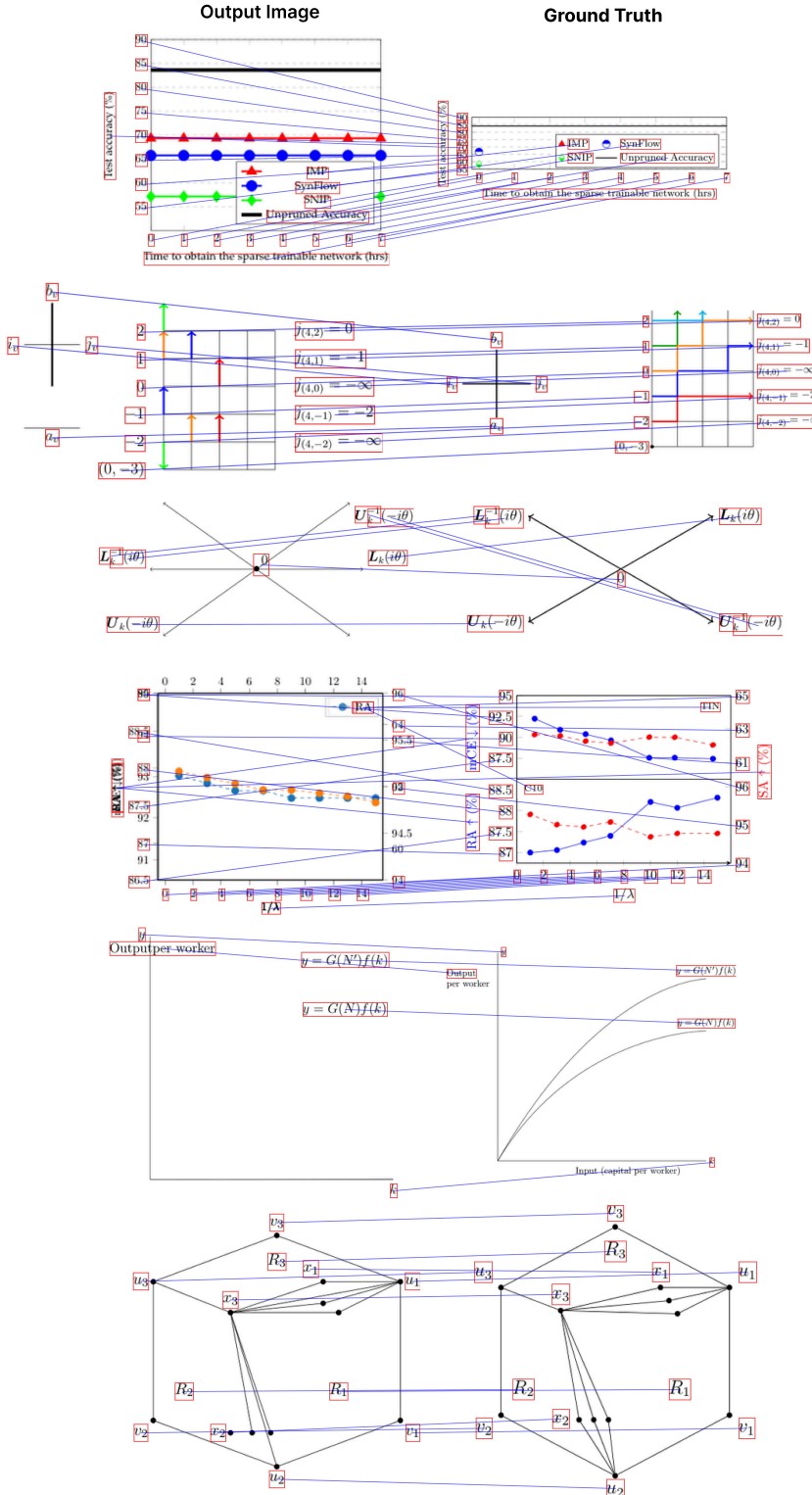

Figure 17: Visualization of the Spatio-Textual match between the ground truth and model output during training. Matched text elements are displayed with bounding boxes and connected by lines; text elements without bounding boxes indicate unmatched.

## E.2 Geometric Similarity Rewarding

The reward function $R_{\text{geom}}$ relies on a type-specific cost function, denoted as $C(e_p, e_g) = \text{CalculateCost}(e_p, e_g)$ in Algorithm 2, to measure the dissimilarity between a predicted element $e_p$ and a ground-truth element $e_g$. This cost is a weighted sum of differences in normalized geometric attributes. Below, we detail the formulation for each primary geometric type.

**Common Notation.** To maintain consistency, we first define several common terms used across the cost functions:

- $c_p, c_g$: The centroids (center points) of elements $e_p$ and $e_g$.
- $D_{\text{canvas}}$: The diagonal of the bounding box enclosing all elements on the page, used for normalization. A small epsilon $\epsilon$ is used to prevent division by zero.
- $d_{pos}(e_p, e_g)$: The normalized Euclidean distance between the centroids of two elements, given by:
$$d_{pos}(e_p, e_g) = \frac{\|c_p - c_g\|_2}{D_{\text{canvas}}}$$
- $d_{size}(e_p, e_g)$: The normalized difference in the primary measure of size $A$ (e.g., area or length) of two elements:
$$d_{size}(e_p, e_g) = \frac{|A_p - A_g|}{\max(A_p, A_g, \epsilon)}$$

**Lines.** A line element is defined by its centroid, length (as its primary size metric, $A_L$), and orientation. The orientation $\theta \in [0, \pi)$ represents the angle of the line. The cost function balances position, length, and angular alignment.

$$C_{\text{line}}(e_p, e_g) = 0.4 \cdot d_{pos}(e_p, e_g) + 0.3 \cdot d_{size}(e_p, e_g) + 0.3 \cdot d_{orient}(e_p, e_g) \tag{6}$$

where the normalized orientation difference $d_{orient}$ is defined as:

$$d_{orient}(e_p, e_g) = \frac{\min(|\theta_p - \theta_g|, \pi - |\theta_p - \theta_g|)}{\pi/2}$$

**Rectangles.** A rectangle is defined by its centroid, area ($A_R$), and aspect ratio $\rho = \text{width/height}$. The cost function considers position, area, and shape similarity via aspect ratio.

$$C_{\text{rectangle}}(e_p, e_g) = 0.4 \cdot d_{pos}(e_p, e_g) + 0.3 \cdot d_{size}(e_p, e_g) + 0.3 \cdot d_{aspect}(e_p, e_g) \tag{7}$$

where the normalized aspect ratio difference $d_{aspect}$ is:

$$d_{aspect}(e_p, e_g) = \frac{|\rho_p - \rho_g|}{\max(\rho_p, \rho_g, \epsilon)}$$

**Circles.** A circle is defined by its center (centroid) $c$ and radius $r$. The cost function heavily weights positional alignment, with a secondary consideration for radius similarity.

$$C_{\text{circle}}(e_p, e_g) = 0.6 \cdot d_{pos}(e_p, e_g) + 0.4 \cdot d_{radius}(e_p, e_g) \tag{8}$$

where the normalized radius difference $d_{radius}$ is:

$$d_{radius}(e_p, e_g) = \frac{|r_p - r_g|}{\max(r_p, r_g, \epsilon)}$$

**Polygons.** A polygon is compared based on its centroid, area ($A_P$), and the number of vertices $V$. This provides a basic measure of its overall shape and complexity.

$$C_{\text{polygon}}(e_p, e_g) = 0.4 \cdot d_{pos}(e_p, e_g) + 0.3 \cdot d_{size}(e_p, e_g) + 0.3 \cdot d_{vertex}(e_p, e_g) \tag{9}$$

where the normalized vertex count difference $d_{vertex}$ is:

$$d_{vertex}(e_p, e_g) = \frac{|V_p - V_g|}{\max(V_p, V_g, \epsilon)}$$

**Curves.** General curves are compared using their centroid, total arc length (as the size metric, $A_C$), and the area of their bounding box ($A_{bbox}$) as a rough proxy for their overall shape and extent.

$$C_{\text{curve}}(e_p, e_g) = 0.4 \cdot d_{pos}(e_p, e_g) + 0.3 \cdot d_{size}(e_p, e_g) + 0.3 \cdot d_{bbox}(e_p, e_g) \tag{10}$$

where $d_{bbox}$ is the normalized difference in bounding box areas:

$$d_{bbox}(e_p, e_g) = \frac{|A_{bbox,p} - A_{bbox,g}|}{\max(A_{bbox,p}, A_{bbox,g}, \epsilon)}$$

---

**Algorithm 2** Geometric Similarity Rewarding

---

1: **Input:** Predicted geometric elements $E_{pred}$.
2: **Input:** Ground-truth geometric elements $E_{gt}$.
3: **Parameter:** Scaling constant $k$.
4: **Output:** Scalar reward score $R_{\text{geom}}$.
5: Initialize set of all matches $M \leftarrow \emptyset$.
                                                    ▷ *Stage 1: Type-wise Optimal Matching*
6: Group elements in $E_{pred}$ and $E_{gt}$ by type (e.g., line, circle, rectangle).
7: Let $\mathcal{T}$ be the set of all unique element types present.
8: **for** each type $t$ in $\mathcal{T}$ **do**
9:     Let $E_p^t$ and $E_g^t$ be the sets of predicted and ground-truth elements of type $t$.
10:     **if** $|E_p^t| > 0$ and $|E_g^t| > 0$ **then**
11:         Construct cost matrix $C^t$ of size $|E_p^t| \times |E_g^t|$.
12:         **for** $i \in \{1, \ldots, |E_p^t|\}, j \in \{1, \ldots, |E_g^t|\}$ **do**
13:             $C_{ij}^t \leftarrow$ CalculateCost($E_p^t[i], E_g^t[j]$).         ▷ Type-specific cost function
14:         **end for**
15:         Matched indices $(P_{idx}, G_{idx}) \leftarrow$ Hungarian($C^t$).
16:         **for** each matched pair of indices $(i, j)$ in $(P_{idx}, G_{idx})$ **do**
17:             $e_p \leftarrow E_p^t[i], e_g \leftarrow E_g^t[j]$.
18:             $M \leftarrow M \cup \{(e_p, e_g)\}$.
19:         **end for**
20:     **end if**
21: **end for**
                                                     ▷ *Stage 2: Final Reward Calculation*
22: $total\_score \leftarrow 0$.
23: **for** each matched pair $(e_p, e_g)$ in $M$ **do**
24:     $cost \leftarrow$ CalculateCost($e_p, e_g$).
25:     $score \leftarrow \exp(-k \cdot cost)$.
26:     $total\_score \leftarrow total\_score + score$.
27: **end for**
28: **if** $\max(|E_{pred}|, |E_{gt}|) > 0$ **then**
29:     $R_{\text{geom}} \leftarrow total\_score / \max(|E_{pred}|, |E_{gt}|)$.
30: **else**
31:     $R_{\text{geom}} \leftarrow 1.0$.
32: **end if**
33: **return** $R_{\text{geom}}$.

---

Figure 18 presents visualizations of the geometric element matches between the ground truth and model output, spanning cases from simple to complex.

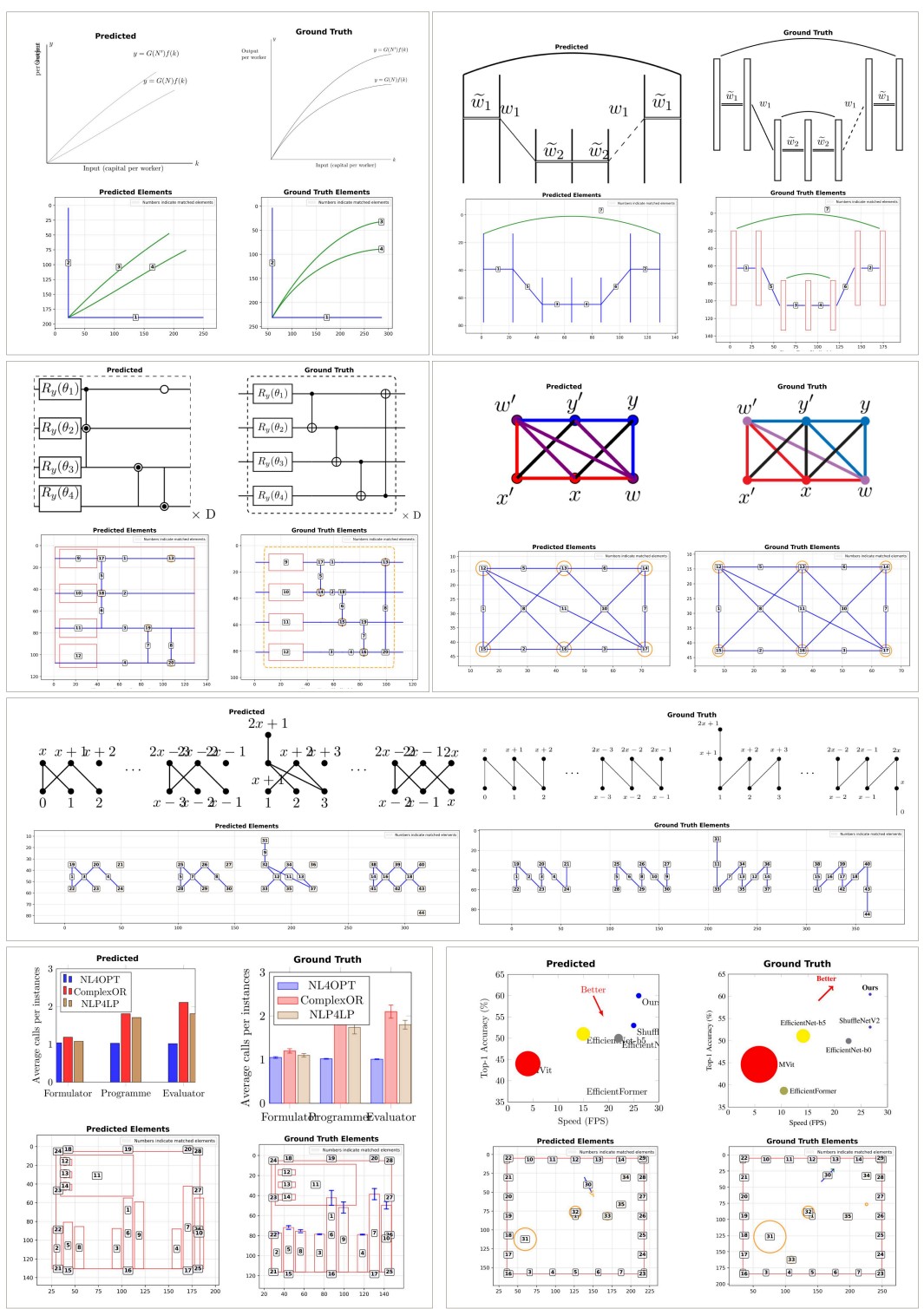

Figure 18: Visualization of geometric element matches. Each sample comprises four sub-images. The top row shows the output (left) and ground truth (right) images. The bottom row displays the extracted geometric elements from the output (left) and ground truth (right). Successfully matched geometric elements are labeled with identical numbers to indicate correspondences between the predicted and ground truth outputs.

## E.3 Training Details

Table 11 and Table 12 show the training settings for the SFT and GRPO training stages, respectively. We freeze the vision encoder and MLP projector during the SFT stage, as this leads to lower training loss variance and smoother convergence compared to unfreezing these components, as shown in Fig. 19.

Table 11: SFT Training Settings.

| Configuration | Setting |
|---|---|
| LLM Training | Unfreeze |
| ViT Training | Freeze |
| MLP Projector Training | Freeze |
| Finetuning Type | Full |
| Cutoff Length | 6144 |
| Learning Rate | $5e^{-5}$ |
| Epochs | 2 |
| Batch Size | 128 |
| Per Device Train Batch Size | 4 |
| Gradient Accumulation Steps | 4 |
| LR Scheduler | Cosine |
| Warmup Ratio | 0.1 |
| GPUs | 8* H100-80G |
| GPU Hours | 2 |

Table 12: GRPO Training Settings.

| Configuration | Setting |
|---|---|
| LLM Training | Unfreeze |
| ViT Training | Unfreeze |
| MLP Projector Training | Unfreeze |
| Max Response Length | 6144 |
| KL Coefficient | 1e-2 |
| Low Clip Threshold | 0.01 |
| High Clip Threshold | 0.99 |
| Rollout Number | 8 |
| Rollout Temperature | 1.0 |
| Rollout Top p | 0.99 |
| Training Steps | 300 |
| Optimizer | AdamW |
| Learning Rate | $1e^{-6}$ |
| Weight Decay | $1e^{-2}$ |
| Per Device Train Batch Size | 4 |
| Rollout Batch Size | 256 |
| Global Batch Size | 128 |
| GPUs | 8* H100-80G |
| GPU Hours | 60 |

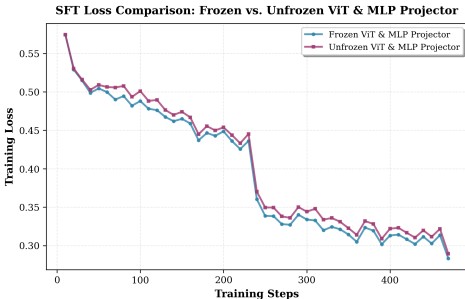

Figure 19: Loss comparison when freeze and unfreeze the vision encoder and the MLP projector during SFT training.

## E.4 OCR Recognition Missing Cases

As shown in Fig. 20, OCR recognition may face incomplete recognition and frequently omits textual content.

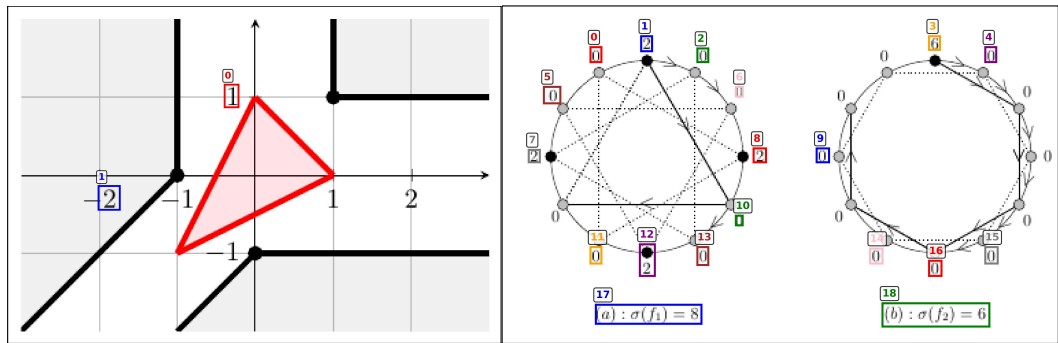

Figure 20: OCR error: missing recognition resulting in omitted textual content.

