# OpenReview forum: "DaVinci: Reinforcing Visual-Structural Syntax in MLLMs for Generalized Scientific Diagram Parsing"
_ICLR.cc/2026/Conference — ICLR 2026 Poster_

### Official Review · Reviewer_zdpU · 2025-10-30

**Soundness:** 3
**Presentation:** 4
**Contribution:** 3
**Rating:** 6
**Confidence:** 4

**Summary:**

This work introduces a new MLLM for synthesizing scientific figures as TikZ graphics programs conditioned on raster images. To facilitate this, the authors introduce a highly curated dataset of 30k TikZ images. Compared to related work, this dataset might seem small, but the automated curation process yields higher-quality training examples that lead to better downstream performance. The authors also introduce a new set of task-specific reward functions that can further improve the model via GRPO. Their final model outperforms both proprietary and open-weight baselines on a wide range of automated metrics.

**Strengths:**

* The central contribution of the work is the dataset and its curation pipeline. Through manual analysis, the authors identify shortcomings of existing datasets and improve their training examples through automated methods. These findings will be useful for future work (such as injecting comments as an "inline" reasoning trace).
* Likewise, the defined reward functions seem effective and scalable and will be useful for future work.
* In general, the paper is well-written and easy to follow.

**Weaknesses:**

* The comparison between DaVinci and DeTikZify is not entirely fair, as DaVinci is based on a stronger base model. This means that conclusions about the effectiveness of the data curation strategy (l.375f) could also be due to this difference.
* Even though the dataset is a central contribution, its creation is not entirely clear. For example, in l.154, the authors mention that they originally collect 360k TikZ snippets. It is not clear at which step these examples are filtered down to only 30k examples.
* While the findings are interesting, there are few technical novelties presented.

**Questions:**

* In l.61, Belouadi et al. use the MCTS algorithm primarily to find outputs with higher perceptual similarity, not to improve the compile success rate.

---

> ### Author Response · Authors · 2025-11-21
> **Response to Reviewer zdpu (1/2)**
>
> We sincerely thank Reviewer zdpu for the thorough review and valuable feedback, which has helped us improve the paper. Below are the detailed revisions addressing your comments; please let us know whether your concerns have been adequately resolved.
>
> ------
>
> **W1**: The comparison between DaVinci and DeTikZify is not entirely fair, as DaVinci is based on a stronger base model. This means that conclusions about the effectiveness of the data curation strategy (l.375f) could also be due to this difference.
>
> **A1**: Thank you for bringing this to our attention. This comparison indeed does not align with our intention to illustrate the high compile rate benefited from our data optimization strategy, because it inadvertently introduced an uncontrolled variable (i.e., the difference in base model capacity).
> - To ensure clarity and focus on the data effectiveness, we have deleted the inappropriate model comparison statement in the original lines 375–376 and redirected the focus to our controlled data ablation study in Section 4.5, which is the core evaluation for our data optimization strategy. We emphasize that the effectiveness of our data optimization is independently validated in this study, where we fix the base model and vary only the data setting. ​​The results in Table 2 show that *reordering* alone increases the compilation success rate by 9.04%↑ over the base data; additional *comment annotation optimization* provides a further gain of 5.72%↑.
> - To further validate the effectiveness of our data curation strategy, we conduct a controlled ablation study on **the public version of DaTikZv3**, which contains **145,366** samples. Applying our filtering and optimization pipeline, we derive a refined subset of **112,177 samples**, denoted as **DaTikZv3-optimized**. We perform supervised fine-tuning (SFT) on both datasets using the same configuration as DaVinci-7B-SFT, with Qwen2.5-VL-7B as the base model.
> The evaluation results on the test set of DaTikZv3-public are presented below, demonstrating that our curation strategy significantly improves Pass@1, increasing from 67.7% to 84.9%, while maintaining or slightly enhancing other key metrics,  with a 22.83% reduction in dataset size.
> | Model | Pass@1 $\uparrow $| CrystalBLEU $\uparrow $ | DreamSIM $ \uparrow $ | SigLIP $ \uparrow $ |  StructuralSIM  $ \uparrow $| MSE $\downarrow $ |
> |-------|----------|-------------|------|--------|------------------|---------------|
> |  Qwen2.5-VL-7B + DatikZV3-public | 67.7% | 3.98 | 82.56 | 93.17 |  **73.85** | **71.13** |
> | Qwen2.5-VL-7B + DatikZV3-optimized | **84.9%** | **7.19** | **82.95** | **93.90** | 72.75 | 72.01 |
>
> We have revised the paper and incorporated the above details in Appendix D.6.
>
> ---------------------
>
> **W2**: Even though the dataset is a central contribution, its creation is not entirely clear. For example, in l.154, the authors mention that they originally collected 360k TikZ snippets. It is not clear at which step these examples are filtered down to only 30k examples.
>
> **A2**: Thank you for pointing this out. We previously left the major clarification of data filtering to Appendix C, leading to the main content being unclear. We have now revised the main content Sect. 3.2 to provide a transparent and self-contained account of our curation pipeline.
>  - We initially source **366,075** runnable TikZ snippets. Through manual inspection, we further identify recurring patterns of low-quality samples to derive a set of heuristic filtering rules, reducing the dataset to **258,421** samples. We then employ Qwen-2.5-VL-32B  to automatically assign semantic class labels and 5-point quality scores to each image, retaining only those with scores of 4 or 5, yielding the final **225,648** high-quality samples.
> - Given that our initial objective is creating a high-quality dataset for efficient cold-start training, we do not apply the expensive LLM-based reordering and comment annotation process to all **225,648** data. We perform stratified sampling based on token length to select a representative subset of 58K samples, since we empirically observe that TikZ program complexity is strongly correlated with sequence length.  This subset is divided into **30K** samples for SFT and 28K for RL.
> - We have also recognized that different research purposes may have varying requirements for data size. To accommodate this diversity, we have applied the LLM-based reordering and comment annotation optimization to the entire **225,648** samples. This complete, high-quality version will be publicly released alongside the smaller 30K subset. Researchers seeking larger-scale training data or alternative sampling strategies can thus access the fully curated corpus and adapt it to their specific needs, eliminating the need for additional financial costs associated with optimizing data with the LLM APIs. Those prioritizing efficiency can directly use the compact 30K version for rapid prototyping and cold-start experiments.

---

> ### Author Response · Authors · 2025-11-21
> **Response to Reviewer zdpu (2/2)**
>
> **W3**: While the findings are interesting, there are few technical novelties presented.
>
> **A3**: We would like to clarify the novel technical insights our work reveals, specifically that the current long-thinking/reasoning style inference-time scaling does not bring expected improvements to the Diagram-to-Code generation tasks. This finding challenges the common assumption that allocating more inference-time computation automatically leads to better performance, especially in multimodal settings where perceptual grounding and structural precision are critical.
>
> However,  we believe the paradigm of inference-time scaling remains promising for diagram parsing and multimodal code generation tasks (e.g., chart-to-code, webpage-to-code), which is underexplored by the current research community.
> We want to share the promising “drawing-with-thinking” paradigm in a supplement discussion paragraph, where generation and reasoning co-evolve through iterative generation, rendered image feedback, and refinement, which could unlock the true potential of scalable inference in these visual code generation domains. This shift from pure language reasoning to active, interactive construction represents, in our view, a meaningful technical direction. Such a paradigm is particularly promising for the generation of drawing languages like TikZ and SVG, where each drawing command could be immediately reflected in the rendered image once compiled.
>
> ---------------------
>
> **Q1**: In l.61, Belouadi et al. use the MCTS algorithm primarily to find outputs with higher perceptual similarity, not to improve the compile success rate.
>
> **A1**: Thank you for this important clarification. We have revised the corresponding text (originally in  l.61) to accurately reflect that Belouadi et al. use the MCTS algorithm to optimize perceptual similarity. We appreciate your careful reading and valuable feedback.

---

> ### Author Response · Authors · 2025-11-25
> **Official Comment by Authors**
>
> Dear Reviewer zdpu,
>
> I hope this message finds you well.
>
> As the discussion period is ongoing and time is running short, we wanted to ensure we have addressed all your concerns satisfactorily. If there are any additional points or feedback you'd like us to consider, please let us know. Your insights are invaluable to us, and we're eager to address any remaining issues to improve our work.
>
> Thank you for your time and effort in reviewing our paper.
>
> Best regards,
>
> Authors

---

> > ### Comment · Reviewer_zdpU · 2025-11-25
> >
> > I thank the authors for their clarifications in the rebuttal and for conducting additional experiments that provide new insights into their method. I agree with other reviewers who highlight the strengths of the data preprocessing pipeline. Therefore, I am maintaining my positive score.

---

> ### Author Response · Authors · 2025-11-30
> **Acknowledgments to Reviewer zdpU**
>
> Dear Reviewer zdpU,
>
> Thank you very much for your positive response. We are very glad that your concerns have been addressed.
>
> We are grateful for your thoughtful feedback and for engaging with our responses throughout the rebuttal process. Thank you again for your time and consideration.
>
> Best regards,
>
> All Authors

---

### Official Review · Reviewer_jdjw · 2025-10-31

**Soundness:** 2
**Presentation:** 2
**Contribution:** 2
**Rating:** 4
**Confidence:** 4

**Summary:**

This paper addresses the timely topic of diagram-to-code generation by proposing a two-stage learning method: a supervised fine-tuning (SFT) "cold start" followed by GRPO post-training. It also explores two overlooked features, code ordering and descriptive comments. However, the paper's key findings lack support from rigorous experimental data, leading to ungrounded claims about the technical method's efficacy. Moreover, the dataset construction and evaluation methodology present a potential risk of bias, indicating that the work requires further improvement before its conclusions can be considered reliable.

**Strengths:**

S1. The proposed two-stage learning approach with SFT + GRPO is a timely fit for the field.

S2. This work draws attention to the often-overlooked features of code ordering and descriptive comments in diagram-to-code generation.

S3. The paper is well written, and the methodology is solid.

**Weaknesses:**

W1. The main concern lies in the data construction process, which contains only 30k samples, whereas previous work (DATIkZ v1: 117k, v2: 360k, v3: 456k) used much larger datasets. Though the authors provide more details in Appendix C, key information is still missing. In line 795, they refer to 225k samples, but after stratified sampling by token length, only 58k remain (line 978), and it is unclear how the final 30k samples (line 149) were selected. Moreover, stratified sampling by token length may introduce bias, making the proposed dataset focus more on the long-tail data of the original dataset.

W2. We encourage the authors to provide a concise and clear description of the data construction process in the main content, showing how 366,075 (line 764) valid pairs were reduced to the final TikZ-30k dataset. It is also necessary to provide comprehensive statistical analyses of the dataset at each step to ensure that no bias is introduced.

W3. Bias in the evaluation method is another critical weakness. In line 338, the authors mention sampling 542 instances from 456,000 instances in DATIkZ v3. This small-scale test set cannot provide statistically convincing results, and the unclear sampling method raises concerns about potential distortion. Moreover, the authors should include evaluation metrics consistent with previous studies for better performance comparison.

W4. The reviewer is curious about the “error-free precise feedback” (line 076); however, the authors only introduce it as “a method” (line 263) without experimental support or technical details.

W5. Format/typos:
- In Figure 1, the Tikz code is changing from “yellow” to “red” while the text is describing “from green to light red” in the top right part.
- In Table 1, did not bold the best score for SigLIP

**Questions:**

See weaknesses

---

> ### Author Response · Authors · 2025-11-21
> **Response to Reviewer jdjw (1/4)**
>
> We sincerely thank Reviewer jdjw for the thorough review and valuable feedback, which has helped us improve the paper. Below are the detailed revisions addressing your comments; please let us know whether your concerns have been adequately resolved.
>
>  ------
>
> **W1**: The main concern lies in the data construction process, which contains only 30k samples, whereas previous work (DATIkZ v1: 117k, v2: 360k, v3: 456k) used much larger datasets. Though the authors provide more details in Appendix C, key information is still missing. In line 795, they refer to 225k samples, but after stratified sampling by token length, only 58k remain (line 978), and it is unclear how the final 30k samples (line 149) were selected. Moreover, stratified sampling by token length may introduce bias, making the proposed dataset focus more on the long-tail data of the original dataset.
>
> **A1**: Thank you for bringing this to our attention. We appreciate the opportunity to clarify the rationale behind our data construction process.
> - **Dataset size differences with prior works**: Our initial collection comprises 366,075 TikZ samples. The discrepancy in initial dataset size compared with DaTikZ v3 (456k) is primarily due to our decision to exclude all data from January 2024 onward to prevent potential data leakage, since the public test set of DaTikZ v3 begins from January 2024. This is also elaborated in the later A3. Then, we identify recurring patterns of low-quality samples to derive a set of filtering rules, followed by employing Qwen-2.5-VL-32B to automatically assign 5-point quality scores to each image, yielding **225,648** samples.
> - **Data subset selection and splitting**:  Given that our initial objective is creating a high-quality dataset for efficient cold-start training, we do not apply the expensive LLM-based reordering and comment annotation process to all 225,648 data. Instead, we perform stratified sampling on the 225k samples based on token length, resulting in a representative subset of 58K samples. The 58K data is further randomly split into **30K** samples for SFT and **28K** for RL, with no overlap between stages. The SFT portion is what we refer to as TikZ30K.
> - **The choice of stratified sampling**: We acknowledge the reviewer’s careful observation that this strategy alters the natural distribution of the data. We appreciate the opportunity to clarify the previously missing details of the bins' separation, where
> we applied stratified sampling based on predefined **unequally spaced** token-count bins: (0, 200], (200, 400], (400, 600], (600, 800], (800, 2000], and (2000, 4096]. This is an intentional design choice aimed at data balancing after observing the token length distribution rather than introducing negative bias. In the context of TikZ generation, code length serves as a strong proxy for visual complexity. Raw data exhibit a heavy "long-tail" distribution (dominated by short code snippets). Random sampling would preserve this imbalance, causing the model to overfit to trivial patterns ("head bias") while under-fitting on complex geometries. By stratifying by token length, we optimize this distribution, ensuring the model receives sufficient supervision signals across diverse difficulty levels during the cold-start stage.
> - We have also recognized that different research purposes may have varying requirements for the size of the final optimized data. To accommodate this diversity, we have applied the LLM-based reordering and comment annotation optimization to the entire **225,648** samples. This complete, high-quality version will be publicly released alongside the smaller 30K subset. Researchers seeking larger-scale training data or alternative sampling strategies can thus access the fully curated corpus and adapt it to their specific needs, eliminating the need for additional financial costs associated with optimizing data with the LLM APIs. Those prioritizing efficiency can directly use the compact 30K version for rapid prototyping and cold-start experiments.
>
> We have revised the paper and incorporated the above details in Sect. 3.2, Appendix C.2, and Appendix C.4.

---

> ### Author Response · Authors · 2025-11-21
> **Response to Reviewer jdjw (2/4)**
>
> **W2**: We encourage the authors to provide a concise and clear description of the data construction process in the main content, showing how 366,075 (line 764) valid pairs were reduced to the final TikZ-30k dataset. It is also necessary to provide comprehensive statistical analyses of the dataset at each step to ensure that no bias is introduced.
>
> **A2**: Thank you for the valuable suggestions.
> - **Data construction process**: Please refer to the above **A1** for the clarification about the data construction process. We have revised the Sect. 3.2 and incorporated the above details clearly.
> - **Comprehensive statistical analyses**: We now provide an overview of how the initial 366,075 valid TikZ–image pairs were progressively refined into the final optimized TikZ-30K dataset, accompanied by statistical analysis at each stage.
>   - Metrics explanation:
>      - Token statistics are reported separately for pure code tokens (excluding comments) and pure comment tokens.
>     - For each stage, we report the minimum and maximum token lengths per sample.
>     - Although we do not apply token-length-based filtering before stratified sampling, as even extremely long samples (up to 1,634,716 tokens) can be syntactically valid and compilable, we exclude samples exceeding 14,096 tokens when computing the average, minimum, and maximum token lengths to reduce distortion from outliers.
>    | Stage | #Samples | Avg. pure code tokens | Avg. comment tokens | Min / Max total tokens | #Samples >14,096 tokens |
> |-------|----------|---------------------------|----------------------|-------------------------|------------------|
> | 0. Initial extraction | 366,075 | 731.15 | 11.89 | 25 / 14,094 | 3269 |
> | 1. Forbidden-keyword filtering (e.g., `includegraphics`, `animate`) | 303,673 | 732.04 | 10.46 | 25 / 14,094 | 2947 |
> | 2. Density-constraint filtering | 298,699 | 696.10 | 9.57 | 25 / 14,087 | 1650 |
> | 3. Filtering via `pdflatex` compilation logs (overfull / truncated / out-of-range) | 258,421 | 625.91 | 8.48 | 25 / 14,087 | 890 |
> | 4. Decimal-rounding refinement | 258,421 | 617.29 | 8.48 | 25 / 14,087 | 888 |
> | 5. LLM-based quality scoring | 225,648 | 617.97 | 8.47 | 25 / 14,087 | 746 |
> | 6. Unequally spaced token-bin sampling to produce 58K | 58,000 | 758.44 | 11.58 | 32 / 4,092 | \ |
> | 7. 30K randomly sampled from 58K | 30,000 | 758.71 | 11.40 | 35 / 4,091 | \ |
> | 8. Final optimized TikZ-30K (code reordering, optimized comments) | 29,859 | 777.85 | 84.56 | 60 / 4,178 | \ |
>    - **Analyses**. We could observe that:
>       -  The stages of `1. Forbidden-keyword filtering`, `4. Decimal-rounding refinement`, `5. LLM-based quality scoring`, and `7. 30K randomly sampled from 58K` show minimal impact on average token counts, indicating that these steps primarily filter or subsample without substantially altering the underlying code complexity distribution. This is as we expected.
>      - `2. Density-constraint filtering` significantly reduces the average code token length (from ~732 to ~696), while ` 3. Filtering via pdflatex compilation logs` causes a further substantial decrease (to ~626 tokens). **These notable changes are compliant with our expectations**, as the removed samples often represent data points that exceed the reasonable scope of the diagram-to-code task. For instance, the density filter removes diagrams with overly dense coordinate data (e.g., attempting to accurately reconstruct positions for over 100 points). Similarly, the pdflatex log filtering eliminates cases leading to truncated images due to LaTeX overfull errors, where content extends beyond the defined page margins and is consequently clipped during compilation. Removing these overly complex or technically flawed samples ensures the dataset's focus on learnable and reproducible diagram-to-code mappings.
>       - `6. Unequally spaced token-bin sampling` intentionally reshapes the token-length distribution, which explains the increase in average pure code tokens from ~618 to 758.
>      - `8. Final optimized TikZ-30K` results in a dramatic increase in comment token length (from ~11.4 to 84.6 on average), as intended.
>
> Overall, no unexpected data distortion was introduced throughout the pipeline; all observed changes in token statistics are consistent with our design intentions, reflecting targeted filtering of overly complex or invalid samples beyond the scope of the diagram-to-code task.
>
> We have revised the paper and incorporated the above details in Appendix C.2.

---

> ### Author Response · Authors · 2025-11-21
> **Response to Reviewer jdjw (3/4)**
>
> **W3**: Bias in the evaluation method is another critical weakness. In line 338, the authors mention sampling 542 instances from 456,000 instances in DATIkZ v3. This small-scale test set cannot provide statistically convincing results, and the unclear sampling method raises concerns about potential distortion. Moreover, the authors should include evaluation metrics consistent with previous studies for better performance comparison.
>
> **A3**: We appreciate this concern and clarify the following points regarding our evaluation setup:
> - **Test set composition**: **We did not perform any ad-hoc sampling from DATikZ v3**. Instead, we use the official test set released by DaTikZ v3, which consists of 542 samples with clear publication date attributions that begin from January 2024. Also, we have strictly cut off our training data in December 2023, preventing any potential distortion (contamination or leakage). We have revised the original line 338 to make it clearer.
> -  **Evaluation metrics consistency**: We adopted suitable standard metrics used in prior work, particularly DeTikZify [1]: cBLEU, TED, DSim, and SSim. We confirm through their public code that the “SSim” reported in DeTikZify corresponds to SigLIP-based similarity (denoted as SigLIP in our study), not the traditional Structural Similarity Index (SSIM).  In contrast, we also compute StructuredSIM, a classic metric, and explicitly denote it as such to avoid confusion.
> - Moreover, to address concerns regarding the scale of the public DaTikZ v3 test set, we construct an additional evaluation set comprising 2,000 samples randomly sampled from 26,909 TikZ code snippets extracted from arXiv papers published between January and October 2025, under licenses permitting redistribution. This part of the data source was not considered in our previous data construction process and was independently collected for the additional evaluation. The results are presented below.
>  | Model | Pass@1↑ | TED↓ | cBLEU↑ | DSIM↑ | SigLIP↑ | SSIM↑ | MSE↓ | LPIPS↓ |
> |--- | --- | --- | --- | --- | --- | --- | --- | --- |
> | **Open Sourced MLLMs** |  |  |  |  |  |  |  |  |
> | Qwen2.5-VL-7B | 78.20 | 54.45 | 6.52 | 77.05 | 90.30 | 74.44 | 82.62 | 27.40 |
> | Qwen2.5-VL-32B | 83.20 | 52.97 | 7.62 | 77.92 | 90.66 | 75.06 | 80.02 | 26.65 |
> | Qwen2.5-VL-72B | 86.30 | 52.71 | 7.66 | 80.52 | 92.12 | 74.52 | 78.09 | 26.17 |
> | GLM-4.5V-106B-A12B | 76.75 | 53.51 | 6.25 | 80.78 | 91.95 | 75.45 | 77.72 | 24.97 |
> | GLM-4.5V-Thinking | 67.10 | 54.92 | 3.59 | 81.15 | 92.43 | **77.13** | 77.58 | 23.72 |
> | **Specialized TikZ-generation MLLMs** |  |  |  |  |  |  |  |  |
> | DetikZify-V2-8B | 86.30 | 51.46 | 10.28 | 85.78 | 93.13 | 76.95 | 67.32 | 22.00 |
> | DiagramAgent-7B | 71.85 | 53.45 | 8.57 | 73.60 | 88.51 | 73.97 | 84.59 | 28.39 |
> | DaVinci-SFT-7B | 89.80 | **50.48** | **11.25** | 85.32 | 93.87 | 76.24 | 69.77 | 22.86 |
> | DaVinci-7B | **99.36** | 52.33 | 9.42 | **86.88** | **94.50** | 75.90 | **61.44** | **20.80** |
>  - This new set appears to be somewhat easier, likely due to its random sampling process, in contrast to the curated, human-selected nature of the DaTikZ v3 test set. The overall performance is consistent with findings on the official DaTikZ v3 test set, such as
>     - Specialized SFT models, DaVinci-SFT-7B and DetikZify-V2-8B, still achieve the best scores on code similarity metrics, i.e., Text Edit Distance (TED) and CrystalBLEU (cBELU).
>     - DaVinci-7B overtakes the best scores across all open-sourced models and specialized TikZ-generation MLLMs.
>     - GLM4.5V with thinking enabled brings a significant decrease in Pass@1, compared with its non-thinking mode.
>
> [1] Belouadi J, Ponzetto S, Eger S. Detikzify: Synthesizing graphics programs for scientific figures and sketches with tikz[J]. Advances in Neural Information Processing Systems, 2024, 37: 85074-85108.
>
> We have revised the paper and incorporated the above details in Appendix D.5.

---

> ### Author Response · Authors · 2025-11-21
> **Response to Reviewer jdjw (4/4)**
>
> **W4**: The reviewer is curious about the "error-free precise feedback" (line 076); however, the authors only introduce it as “a method” (line 263) without experimental support or technical details.
>
> **A4**: We sincerely thank the reviewer for this observation. We have clarified the meaning of "error-free precise feedback" in the revised manuscript (original line 076) by explicitly defining it as: "extract textual and geometric primitives in an extraction-error-free manner to provide precise feedback".
>
>  - **Conventional approaches for element extraction**. They rely on OCR for text recognition and heuristic-based image processing or object detection for geometric elements. These methods are prone to errors such as character misrecognition (especially for mathematical notation), bounding box misalignment, and font ambiguity, limiting the reliability of using their results for constructing further feedback signals.
>
> - **Error-free extraction from vectorization representations**. We exploit the fact that TikZ generates PDFs in which both textual and geometric elements are stored as structured, machine-readable vector objects, complete with exact coordinates, font metadata, and drawing commands. By directly parsing these vector elements using the PDF-processing library PyMuPDF, we bypass visual recognition entirely. This grants us ground-truth access to all textual content, geometric primitives (e.g., lines, shapes, paths), and their precise spatial attributes (e.g., coordinates, bounding boxes), all without reconstruction error.
> We have carefully revised the Spatio-Textual Similarity Reward (originally line 263) for clarity.
> -  Additionally, we have added visualizations to show the extracted element correspondences derived by the proposed matching algorithms. Specifically, we add
>    - Figure 17 (Appendix E.1) shows correspondences between text elements.
>    - Figure 18 (Appendix E.2) illustrates geometric element matches, covering a spectrum of cases from simple primitives to complex composite structures.

---

> ### Author Response · Authors · 2025-11-25
> **Official Comment by Authors**
>
> Dear Reviewer jdjw,
>
> I hope this message finds you well.
>
> As the discussion period is ongoing and time is running short, we wanted to ensure we have addressed all your concerns satisfactorily. If there are any additional points or feedback you'd like us to consider, please let us know. Your insights are invaluable to us, and we're eager to address any remaining issues to improve our work.
>
> Thank you for your time and effort in reviewing our paper.
>
> Best regards,
>
> Authors

---

> ### Comment · Reviewer_jdjw · 2025-11-25
>
> Dear authors,
>
> Thank you for your detailed and comprehensive response. It addressed all of my concerns, and I have raised my score accordingly.
>
> Best.
>
> Reviewer jdjw

---

> ### Author Response · Authors · 2025-11-30
> **Acknowledgments to Reviewer jdjw**
>
> Dear Reviewer jdjw,
>
> Thank you very much for your positive response. We are very glad that **all of your concerns have been addressed**, and we sincerely **appreciate your decision to raise the score**.
>
> We are grateful for your thoughtful feedback and for engaging with our responses throughout the rebuttal process. Thank you again for your time and consideration.
>
> Best regards,
>
> All Authors

---

### Official Review · Reviewer_1soJ · 2025-10-31

**Soundness:** 3
**Presentation:** 3
**Contribution:** 3
**Rating:** 6
**Confidence:** 4

**Summary:**

This work addresses the problem of parsing raster-based scientific figures into TikZ code. The authors introduce TikZ30K, a high-quality dataset constructed through extensive preprocessing using large language models (LLMs) and vision-language models (VLMs). The data curation pipeline includes low-quality data removal via VLMs, code reordering via LLMs, and comment injection to enhance planning capabilities. A Qwen2.5-VL multimodal LLM is fine-tuned using a two-step training pipeline consisting of supervised fine-tuning (SFT) followed by reinforcement learning (RL). The authors design novel reward functions evaluating textual information, geometric elements, image fidelity, and rendering success rate. The trained model is evaluated on a range of proprietary and open-source MLLMs, demonstrating that the SFT+RL approach outperforms baselines such as GPT-5 on automatic metrics.

**Strengths:**

- The data preprocessing pipeline is novel and intuitive. The use of VLMs for labeling and filtering, LLMs for code reordering, and comment injection for planning is well-motivated. Ablation studies confirm that reordering improves performance.
- The reward design is comprehensive, capturing textual, geometric, and rendering aspects. The inclusion of ablations on different reward settings is appreciated.
- The analysis provides interesting insights, particularly regarding reasoning vs. non-reasoning behavior and the observation that code similarity decreases while performance improves after RL. Evaluation includes both proprietary and open-source MLLMs.

**Weaknesses:**

- It would be helpful to include a code efficiency metric, such as average token or character length, to better understand the complexity and verbosity of the generated TikZ code.
- There is no information about the data timeframe or sourcing, which raises potential concerns about data contamination or leakage.
- The lack of human evaluation is a major limitation. Relying solely on automatic metrics makes it difficult to assess the real-world quality and fidelity of generated figures. Even a small-scale human evaluation would substantially increase the credibility of the results and help correlate human judgments with automatic metrics.

**Questions:**

- What is the exact difference between Reorder30K and TikZ30K? Is it simply the inclusion of comments introduced during data curation?
- For DaTikZ-V3, is the preprocessing pipeline the same as TikZ30K? Also, is TikZ30K a subset of DaTikZ-V3, or was it sourced independently? Clarifying the dataset relationships and data collection process in the main paper would improve transparency.

---

> ### Author Response · Authors · 2025-11-21
> **Response to Reviewer 1soj (1/3)**
>
> We sincerely thank Reviewer 1soj for the thorough review and valuable feedback, which has helped us improve the paper. Below are the detailed revisions addressing your comments; please let us know whether your concerns have been adequately resolved.
>
> **W1**: It would be helpful to include a code efficiency metric, such as average token or character length, to better understand the complexity and verbosity of the generated TikZ code.
>
> **A1**: Thank you for your valuable advice. The table below presents the 10% winsorized mean token and character length, following the previous study, Detikzify [1], to maintain consistency. Winsorization replaces the top and bottom 10% of values with the 10th and 90th percentiles, reducing outlier influence while keeping the sample size.
>
> We observe that:
> - The most capable models in Image-to-TikZ generation, Gemini-2.5-Pro-Thinking (1640 chars, 914 tokens) and DaVinci-7B (1618 chars, 983 tokens), exhibit similar code efficiency. Notably, the other three proprietary MLLMs all produce shorter outputs, indicating higher code efficiency, with GPT-5-Default (1081 chars, 604 tokens) achieving the most concise results.
> - Among specialized TikZ-generation MLLMs, DaVinci-7B achieves the highest code efficiency and is similar to its base model, Qwen2.5-VL-7B.
> - **Code efficiency improves with model scale**, as demonstrated by the Qwen2.5-VL series (7B, 32B,72B). This trend suggests that scaling up model size may enhance code efficiency.
>  | Models                 | Avg_Char_Length_Winsorized | Avg_Token_Count_Winsorized |
> |----------------------------|----------------------------:|----------------------------:|
> | **Proprietary MLLMs**      |                            |                            |
> | Gemini2.5-Pro-Thinking     | 1640.07                    | 914.24                     |
> | GPT-5-Default              | **1081.45**                    | **603.93**                     |
> | Claude-Sonnet-4            | 1320.93                    | 778.69                     |
> | Claude-Sonnet-4-thinking   | 1243.53                    | 727.16                     |
> | **Open-Sourced MLLMs**     |                            |                            |
> | Qwen2.5-VL-7B              | 1687.20                    | 1132.25                    |
> | Qwen2.5-VL-32B             | 1186.80                    | 653.85                     |
> | Qwen2.5-VL-72B             | 1107.85                    | 615.94                     |
> | **Specialized TikZ-generation MLLMs** |                            |                            |
> | DetikZify-V2-8B            | 2357.65                    | 1337.33                    |
> | DiagramAgent-7B            | 1955.69                    | 1191.66                    |
> | *DaVinci-SFT-7B*             | 1722.87                    | 984.38                     |
> | *DaVinci-7B*                 | 1618.04                    | 983.01                     |
>
> We have revised the paper and incorporated the above results and analysis details in Appendix D.3 Code Efficiency Metric due to the space limitation of the main content.
>
> [1] Belouadi J, Ponzetto S, Eger S. Detikzify: Synthesizing graphics programs for scientific figures and sketches with tikz[J]. Advances in Neural Information Processing Systems, 2024, 37: 85074-85108.
>
> ----------------------
>
> **W2**: There is no information about the data timeframe or sourcing, which raises potential concerns about data contamination or leakage.
>
> **A2**: We appreciate this important concern. We have now added detailed information regarding the data timeframe and temporal separation at the beginning of the Data Collection in the revised manuscript. Specifically, our training and test sets are **strictly temporally separated**:
> - All training data for our models is drawn exclusively from sources dated **by December 2023**.
> - The test set we used, i.e., the official test set of DaTikZ-V3, consists of TikZ snippets collected from sources published **from January 2024 onward**.
>
> We have revised the paper and incorporated the timeframe details in Section 3.2 Data Collection.

---

> ### Author Response · Authors · 2025-11-21
> **Response to Reviewer 1soj (2/3)**
>
> **W3**:  The lack of human evaluation is a major limitation. Relying solely on automatic metrics makes it difficult to assess the real-world quality and fidelity of generated figures. Even a small-scale human evaluation would substantially increase the credibility of the results and help correlate human judgments with automatic metrics.
>
> **A3**:  Thank you for the suggestion. We perform human evaluation studies using *Best-Worst Scaling* (BWS), evaluating 100 items randomly sampled from the test set, in line with prior work, Detikzify. For each reference figure, **four** output images generated by different models are displayed side-by-side. Six human evaluators are tasked to select the single best and single worst output. These pairwise selections are aggregated and converted into normalized scores ranging from -1 (worst) to 1 (best), computed as the difference between the proportion of times a model’s output is chosen as best $p_{best}$ and the proportion chosen as worst $p_{worst}$.
> Specifically,  we conducted two groups of human evaluations to ensure comprehensiveness:
> - Group 1: DaVinci-7B and the top three non-proprietary models according to automatic metric evaluation, i.e., DetikZify-V2-8B, Qwen2.5-VL-72B, and DaVinci-SFT-7B;
> - Group 2: DaVinci-7B and three proprietary models, i.e., Gemini2.5-Pro-Thinking, GPT5-Default, and Claude-Sonnet-4-Thinking.
>
> Tables 1  and 2 present the two groups of human evaluation results, which generally align with the automatic evaluation metrics.
> In the non-proprietary model group, DaVinci-7B ( $\mu=0.365$ ) achieves the highest mean score, outperforming all models.
> DaVinci-SFT-7B and Detikzify-V2-8B exhibit comparable performance, while Qwen2.5-VL-72B performs the worst, highlighting the potential improvement by scaling TikZ-specialized models.
> Consistent with automatic evaluation, Gemini-2.5-Pro-Thinking ($\mu=0.50$) significantly outperforms all other models in both groups, showing its superior capability in this task.
> Notably, DaVinci-7B also demonstrates stronger performance than GPT-5-Default and Claude-Sonnet-4-thinking in terms of $p_{\text{best}}$ and $p_{\text{worst}}$.
> - **Table 1**: Human evaluation results of Group 1.
>    | Model               | $p_{\text{best}}$ | $p_{\text{worst}}$ | score  | std   |
> |---------------------|-------------------|--------------------|--------|-------|
> | Qwen2.5-VL-72B      | 0.13              | 0.39               | -0.26  | 0.06  |
> | DetikZify-V2-8B     | 0.25         | 0.30               | -0.05  | 0.06  |
> | DaVinci-SFT-7B      | 0.16              | 0.21           | -0.05  | 0.04 |
> | DaVinci-7B          | **0.47**          | **0.11**           | **0.36** | **0.03** |
>
> - **Table 2**: Human evaluation results of Group 2.
>   | Model                      | $p_{\text{best}}$ | $p_{\text{worst}}$ | score  | std   |
> |----------------------------|-------------------|--------------------|--------|-------|
> | Gemini-2.5-Pro-Thinking    | **0.58**          | **0.08**           | **0.50** | 0.10  |
> | GPT-5-Default              | 0.13              | 0.26               | -0.13  | **0.03** |
> | Claude-Sonnet-4-Thinking   | 0.10              | 0.45               | -0.35  | 0.05|
> | DaVinci-7B                 | 0.20          | 0.21          | -0.01 | 0.06  |
>
> We have revised the paper and incorporated the above details in Section 4.4 (Human Evaluation).

---

> ### Author Response · Authors · 2025-11-21
> **Response to Reviewer 1soj (3/3)**
>
> **Q1**: What is the exact difference between Reorder30K and TikZ30K? Is it simply the inclusion of comments introduced during data curation?
>
> **A1**: Yes, the difference lies in the optimizations applied: Reorder30K incorporates only ordering optimization, while TikZ30K includes both ordering and comment optimization. Our ablation study isolates the individual contributions of these two optimization components.
> The results of the ablation study show that the two optimization aspects bring improvements.
>
> We have revised the relevant sentences in Section 4.5 Ablation Study to make the difference clearer for readers. Thank you for the valuable reading feedback.
>
> ----------------------
>
> **Q2**: For DaTikZ-V3, is the preprocessing pipeline the same as TikZ30K? Also, is TikZ30K a subset of DaTikZ-V3, or was it sourced independently? Clarifying the dataset relationships and data collection process in the main paper would improve transparency.
>
>
> **A2**:  Thank you for bringing this to our attention,
> - **Preprocessing pipeline is different**: While both datasets perform basic compile checks and white image detection (identifying TikZ code that compiles to blank output), our datasets incorporate additional quality control measures, including rule-based filtering, LLM-as-a-judge quality assessment, and normalization (e.g., redundant package removal). Furthermore, our pipeline uniquely addresses the order and comment optimization as a central contribution, which is not present in TikZ30K's preprocessing.
> - **Dataset relationship**: The original code of TikZ30K is not a subset of the public version of DaTikZ-V3, but maybe largely overlaps with the in-house version of DaTikZ-V3. Our data pipeline begins with reproducing the collection process of the DaTikZ series, since the non-distributable licenses of some underlying data sources from which the DaTikZ datasets were originally compiled.  The public version of DaTikZ-V3 contains 145K examples, compared to 440K in their in-house version. Therefore, we re-collect data from the same sources, while getting 366K. The discrepancy in volume is primarily due to our decision to exclude all data from January 2024 onward to prevent potential data leakage.
>
> We have revised the paper to clarify the dataset relationship and data collection process at the beginning of Section 3.2.

---

> ### Author Response · Authors · 2025-11-25
> **Official Comment by Authors**
>
> Dear Reviewer 1soj,
>
> I hope this message finds you well.
>
> As the discussion period is ongoing and time is running short, we wanted to ensure we have addressed all your concerns satisfactorily. If there are any additional points or feedback you'd like us to consider, please let us know. Your insights are invaluable to us, and we're eager to address any remaining issues to improve our work.
>
> Thank you for your time and effort in reviewing our paper.
>
> Best regards,
>
> Authors

---

> > ### Comment · Reviewer_1soJ · 2025-11-25
> >
> > Dear Authors,
> >
> > thanks a lot for your additions. They are generally good, and I would like to keep my positive assessment for your work.
> >
> > Some issues that could further be improved are the following:
> >
> > 1) While your test data is split off temporally from your training data, the models may still have seen the test data distribution.
> > 2) for the human evaluation, there is no indication of the demographics of the annotators, the annotation cost and the agreement among annotators?
> > 3) What is the agreement of human and automatic evaluation?
> >
> > Btw. it's human evaluation, not "humane evaluation"
> >
> > Best

---

> ### Author Response · Authors · 2025-11-30
> **Response to Further Questions from Reviewer 1soj (1/2)**
>
> Thank you for acknowledging our previous replies and revisions.  Below are the detailed revisions regarding additional questions you raised, which further help us improve the paper quality.
>
> -----------
>
> **Further Q1**: While your test data is split off temporally from your training data, the models may still have seen the test data distribution.
>
> **Further A1**: We appreciate the reviewer's rigorous scrutiny regarding data leakage. We address this concern from two levels: sample-level decontamination and distribution-level consistency.
> - **Strict Decontamination (Sample Level)**: Beyond the temporal split, we implemented rigorous deduplication to ensure no exact memorization occurs. Following standard practices [1, 2], we employed N-gram overlap checks (for code) and addtional Mean Square Errors similarity checks (for images) to filter out potential duplicates. These metrics confirm that our test samples are unseen during training.
> - **Inherent Nature of Distribution (Distribution Level)**: We respectfully argue that while sample isolation is mandatory, complete distributional isolation is often neither feasible nor desirable in the context of foundation models and graphics drawing language.
>     - Feasibility: Foundation models are pre-trained on internet-scale data; thus, the general "distribution" of TikZ patterns is inevitably seen.
>     - Learnability & TikZ Nature: TikZ is a rigorous graphic drawing language governed by strict syntax and compilation rules (e.g., coordinate systems, path commands). If the test distribution were completely disjoint from the training distribution (i.e., varying syntax or unknown libraries), the model would fail to generate executable code. The model must adhere to the fixed "alphabet" of TikZ keywords and drawing command usages to produce executable code.
>
> -----------
>
> **Further Q2**: For the human evaluation, there is no indication of the demographics of the annotators, the annotation cost, and the agreement among annotators?
>
> **Further A2**:  We sincerely thank the reviewer for raising this point and provide the following details:
>  - **Demographics and Annotation Cost**: Participants are graduate students aged from 23 to 29 years ($M$ = 25.83, $SD$ = 2.23), with 2 males and 4 females. Each participant received compensation equivalent to US$15 for their participation.
> - **Agreement among Annotators**: In line with DeTikZify [1], we measure agreement among annotators using split-half reliability (SHR) [3]. For each study, we randomly split annotators into two groups.
> For each image, we aggregate the Best-Worst Scaling annotations within each group to obtain vectors of model scores, compute the Spearman rank correlation $\rho$ between the groups, and report the mean correlation across all images.
> The resulting SHR values of $\rho$ = 0.7227 for Study 1 and $\rho$ = 0.7878 for Study 2 indicate strong inter-annotator agreement, supporting the reliability of our human evaluation results.
>
> We have revised the paper and incorporated the above details in Sect. 4.4 and Appendix D.4.2.
>
> [1] Belouadi J, Ponzetto S, Eger S. Detikzify: Synthesizing graphics programs for scientific figures and sketches with tikz[J]. Advances in Neural Information Processing Systems, 2024, 37: 85074-85108.
>
> [2] Belouadi J, Ilg E, Keuper M, et al. Tikzero: Zero-shot text-guided graphics program synthesis[C]//Proceedings of the IEEE/CVF International Conference on Computer Vision. 2025: 17793-17806.
>
> [3] Kiritchenko S, Mohammad S. Best-worst scaling more reliable than rating scales: A case study on sentiment intensity annotation[C]//Proceedings of the 55th Annual Meeting of the Association for Computational Linguistics (Volume 2: Short Papers). 2017: 465-470.

---

> ### Author Response · Authors · 2025-11-30
> **Response to Further Questions from Reviewer 1soj (2/2)**
>
> **Further Q3**: What is the agreement of human and automatic evaluation?
>
> **Further A3**:   We sincerely thank the reviewer for raising this question and add the following analysis:
> - **Agreement in Overall Conclusions**.
>   - In Group 1, DaVinci-7B achieves the highest mean score, outperforming all models. DaVinci-SFT-7B and Detikzify-V2-8B exhibit comparable performance;
>   - In Group 2, Gemini-2.5-Pro-Thinking significantly outperforms all other models, showing its superior capability in this task. Notably, DaVinci-7B also demonstrates stronger performance than GPT-5-Default and Claude-Sonnet-4-thinking.
>
>   These qualitative trends are consistent with the conclusions drawn from our automatic evaluation results: *DaVinci-7B overtakes the best scores across all open-sourced models,  and surpasses the leading commercial models like GPT-5 and Claude-Sonnet-4*.
> - **Agreement with Specific Automatic Metrics**.
>   - Following previous work, DetikZify [1], we derive model rankings based on image similarity metrics and compute Spearman rank correlation coefficients between these rankings and those obtained from aggregated human annotations.
> We observe strong agreement between automatic metrics and human judgments, confirming the reliability of these metrics for evaluating overall model performance. The correlations are summarized in the table below:
>   | Metric        | Group 1 | Group 2 |  Mean |
> |---------------|----------:|----------:|------------:|
> | **LPIPS**     | 0.949     | 1.000     | 0.974   |
> | **DSIM**      | 0.949     | 1.000     | 0.974   |
> | **SigLIP**    | 0.949     | 1.000     | 0.974   |
> | **MSE**       | 0.949     | 0.800     | 0.874   |
> | **SSIM**      | 0.633     | 0.800     | 0.716   |
>
>   - The consistently high Spearman correlations (particularly for LPIPS, DSIM, and SigLIP) indicate that these metrics are effective in the evaluation setting, which is reasonable since they are widely used image similarity metrics.
>
> We have revised the paper and incorporated the above details in Appendix D.4.2.
>
> [1] Belouadi J, Ponzetto S, Eger S. Detikzify: Synthesizing graphics programs for scientific figures and sketches with tikz[J]. Advances in Neural Information Processing Systems, 2024, 37: 85074-85108.
>
> ---------
>
> **Further Comments**: Btw. it's human evaluation, not "humane evaluation"
>
> **Answer**: Thank you for your careful reading. We have corrected the typo from “humane evaluation” to “human evaluation.”

---

### Official Review · Reviewer_modw · 2025-11-01

**Soundness:** 2
**Presentation:** 3
**Contribution:** 3
**Rating:** 6
**Confidence:** 4

**Summary:**

This paper presents a two-stage learning framework for diagram generation using TikZ, consisting of supervised fine-tuning (SFT) followed by reinforcement learning (RL).
In the first stage, the authors investigate the effects of drawing order and comment placement, aspects that have not been thoroughly examined in previous studies.
In the second stage, they introduce a composite reward function in the RL phase that incorporates multiple factors, including image quality, code correctness, and structural consistency, and demonstrate its effectiveness through experimental results.

**Strengths:**

- The authors analyze previously overlooked properties of the data itself—specifically, the order in which data are generated and the use of comments within the code—and demonstrate their positive effects.
- They further propose a method for defining the RL reward that incorporates text alignment, bounding-box consistency, and element matching directly extracted from the vector images.

**Weaknesses:**

- Although a new dataset has been constructed, there is no description of its license information, making the details of the dataset’s licensing unclear.
- The evaluation relies solely on automatic metrics, and no human evaluation is conducted. Since the proposed automatic metrics partially overlap with the reward function used in RL, verifying through human evaluation whether the improvement is genuinely meaningful—or merely a result of metric hacking—would greatly enhance the credibility and reliability of the results.

**Questions:**

- In Section E.3 of the paper, which describes the training details, it is mentioned that the ViT and MLP projector are frozen during the SFT stage but unfrozen during the GRPO stage. Did you experiment with both configurations—freezing and unfreezing—in the GRPO stage, and find that unfreezing these components led to better performance?
- Will the model, code, and dataset be released to the public?
- The dataset is referred to as TikZ30K, but according to Appendix C.4, there are 58,000 training samples. Were the data further filtered during the optimization or post-verification steps? Does this mean that the final dataset actually consists of 30,000 samples?
- The code augmentation process does not seem trivial for an LLM. The paper mentions that the procedure was repeated when rendering failed or when the rendered image was dissimilar to the original. On average, how many attempts were required per sample to obtain a valid code conversion? Moreover, were all samples successfully converted into valid code, or were some data filtered out due to unsuccessful conversions?
- Could you please clarify which specific tools or libraries were used to extract the elements from the PDF files when computing the geometric similarity reward?
- In the computation of the geometric similarity reward, I assume that matching the corresponding elements is not a trivial task. Could you please elaborate on how reliable the matching results are when inspected by humans? In other words, to what extent does the proposed algorithm produce visually or semantically reasonable correspondences between elements?

---

> ### Author Response · Authors · 2025-11-21
> **Response to Reviewer modw (1/4)**
>
> We sincerely thank Reviewer modw for the thorough review and valuable feedback, which has helped us improve the paper. Below are the detailed revisions addressing your comments; please let us know whether your concerns have been adequately resolved.
>
> **W1**: Although a new dataset has been constructed, there is no description of its license information, making the details of the dataset’s licensing unclear.
>
> **A1**: Thank you for pointing this out. Our data release will strictly adhere to the licensing terms of the sources from which we extract the TikZ code snippets. The licenses involved include the Creative Commons Attribution Licenses, the MIT License, the GNU Free Documentation License,  and the arXiv.org perpetual, non-exclusive license. Specifically:
> - For code snippets sourced from permissively licensed platforms that allow redistribution, we directly release the original and optimized code, including the Creative Commons Attribution Licenses, the GNU Free Documentation License, or the MIT license for websites, GitHub repositories, and part of arXiv sources, with clear data source attribution per code snippet.
> - For content subject to redistribution restrictions (notably, the original arXiv submissions under its non-exclusive license), we do not redistribute the raw source. Instead, we will provide diff files and fully documented reproducible scripts that allow researchers to locally reconstruct the optimized versions from publicly available arXiv sources, ensuring full compliance with copyright terms, without imposing additional time or financial burdens on researchers.
>
> We have revised the paper and included the above license information titled "Data Release and License Information" at the end of the main content.

---

> ### Author Response · Authors · 2025-11-21
> **Response to Reviewer modw (2/4)**
>
> **W2**:  The evaluation relies solely on automatic metrics, and no human evaluation is conducted. Since the proposed automatic metrics partially overlap with the reward function used in RL, verifying through human evaluation whether the improvement is genuinely meaningful—or merely a result of metric hacking—would greatly enhance the credibility and reliability of the results
>
> **A2**:  We appreciate this important concern. To address it, we have conducted complementary human evaluation studies.
> - **Study Settings**: We use *Best-Worst Scaling* (BWS) [2], evaluating 100 items randomly sampled from the test set, in line with prior work, Detikzify [1]. For each reference figure, **four** output images generated by different models are displayed side-by-side. Six human evaluators are tasked to select the single best and single worst output. These pairwise selections are aggregated and converted into normalized scores ranging from -1 (worst) to 1 (best), computed as the difference between the proportion of times a model’s output is chosen as best $p_{best}$ and the proportion chosen as worst $p_{worst}$.
> Specifically,  we conducted two groups of human evaluations to ensure comprehensiveness:
>
>   - Group 1: DaVinci-7B and the top three non-proprietary models according to automatic metric evaluation, i.e., DetikZify-V2-8B, Qwen2.5-VL-72B, and DaVinci-SFT-7B;
>    - Group 2: DaVinci-7B and three proprietary models, i.e., Gemini2.5-Pro-Thinking, GPT5-Default, and Claude-Sonnet-4-Thinking.
>
> - **Results**:
> Tables 1  and 2 present the two groups of human evaluation results, which generally align with the automatic evaluation metrics.
> In the non-proprietary model group, DaVinci-7B ( $\mu=0.365$ ) achieves the highest mean score, outperforming all models.
> DaVinci-SFT-7B and Detikzify-V2-8B exhibit comparable performance, while Qwen2.5-VL-72B performs the worst, highlighting the potential improvement by scaling TikZ-specialized models.
> Consistent with automatic evaluation, Gemini-2.5-Pro-Thinking ($\mu=0.50$) significantly outperforms all other models in both groups, showing its superior capability in this task.
> Notably, DaVinci-7B also demonstrates stronger performance than GPT-5-Default and Claude-Sonnet-4-thinking in terms of $p_{\text{best}}$ and $p_{\text{worst}}$.
>   - Table 1: Human evaluation results of Group 1.
>    | Model               | $p_{\text{best}}$ | $p_{\text{worst}}$ | score  | std   |
> |---------------------|-------------------|--------------------|--------|-------|
> | Qwen2.5-VL-72B      | 0.13              | 0.39               | -0.26  | 0.06  |
> | DetikZify-V2-8B     | 0.25         | 0.30               | -0.05  | 0.06  |
> | DaVinci-SFT-7B      | 0.16              | 0.21           | -0.05  | 0.04 |
> | DaVinci-7B          | **0.47**          | **0.11**           | **0.36** | **0.03** |
>
>    - Table 2: Human evaluation results of Group 2.
>   | Model                      | $p_{\text{best}}$ | $p_{\text{worst}}$ | score  | std   |
>   |----------------------------|-------------------|--------------------|--------|-------|
>   | Gemini-2.5-Pro-Thinking    | **0.58**          | **0.08**           | **0.50** | 0.10  |
>   | GPT-5-Default              | 0.13              | 0.26               | -0.13  | **0.03** |
>   | Claude-Sonnet-4-Thinking   | 0.10              | 0.45               | -0.35  | 0.05|
>   | DaVinci-7B                 | 0.20          | 0.21          | -0.01 | 0.06  |
>
> - **Agreement among Annotators**: In line with DeTikZify [1], we measure agreement among annotators using split-half reliability (SHR) [2]. For each study, we randomly split annotators into two groups.
> For each image, we aggregate the Best-Worst Scaling annotations within each group to obtain vectors of model scores, compute the Spearman rank correlation $\rho$ between the groups, and report the mean correlation across all images.
> The resulting SHR values of $\rho$ = 0.7227 for Study 1 and $\rho$ = 0.7878 for Study 2 indicate strong inter-annotator agreement, supporting the reliability of our human evaluation results.
>
>  - **Demographics and Annotation Cost**: Participants are graduate students aged from 23 to 29 years ($M$ = 25.83, $SD$ = 2.23), with 2 males and 4 females. Each participant received compensation equivalent to US$15 for their participation.
>
> We have revised the paper and incorporated the above details in Section 4.4 (Human Evaluation).
>
> [1] Belouadi J, Ponzetto S, Eger S. Detikzify: Synthesizing graphics programs for scientific figures and sketches with tikz[J]. Advances in Neural Information Processing Systems, 2024, 37: 85074-85108.
>
> [2] Kiritchenko S, Mohammad S. Best-worst scaling more reliable than rating scales: A case study on sentiment intensity annotation[C]//Proceedings of the 55th Annual Meeting of the Association for Computational Linguistics (Volume 2: Short Papers). 2017: 465-470.

---

> ### Author Response · Authors · 2025-11-21
> **Response to Reviewer modw (3/4)**
>
> **Q1**: In Section E.3 of the paper, which describes the training details, it is mentioned that the ViT and MLP projector are frozen during the SFT stage but unfrozen during the GRPO stage. Did you experiment with both configurations—freezing and unfreezing—in the GRPO stage, and find that unfreezing these components led to better performance?
>
> **A1**: Thank you for your careful reading.
> - For the SFT stage, we conducted experiments to explore the effect of freezing and unfreezing.  We find that freezing the vision encoder and MLP projector during the SFT stage leads to lower training loss variance and smoother convergence compared to unfreezing these components. Since unfreezing them in the SFT stage does not bring benefits, freezing them is a more reliable choice to provide a stable initialization for subsequent RL training.
>
> - For the GRPO stage, we did not conduct an extra ablation between freezing and unfreezing the ViT and MLP projector due to computational costs. Our decision to unfreeze them was motivated by the fact that GRPO incorporates KL divergence regularization, which constrains policy updates and reduces the risk of instability of the vision representation. Also, unfreezing may provide more flexibility for RL-driven representation refinement.
>
> We have added Figure 19 in Section E.3, which compares the supervised fine-tuning (SFT) training losses under unfrozen and frozen ViT and MLP settings.
>
> ----------------------
>
> **Q2**: Will the model, code, and dataset be released to the public?
>
> **A2**: Yes. Our model, code, and dataset will be released to the public. We also elaborate on our efforts to improve the soundness and contribution of the dataset release in the following A3.
>
> ----------------------
>
> **Q3**: The dataset is referred to as TikZ30K, but according to Appendix C.4, there are 58,000 training samples. Were the data further filtered during the optimization or post-verification steps? Does this mean that the final dataset actually consists of 30,000 samples.
>
> **A3**: Thank you for pointing out this ambiguity regarding the transition from 58,000 samples to the final TikZ30K. This difference does not arise from optimization or post-verification steps. **58,000** samples are initially divided into two splits: 30,000 samples for supervised fine-tuning (SFT) and 28,000 samples for reinforcement learning (RL). This separation ensures no overlap between training stages, thereby avoiding the degradation of the model’s exploration capability in reinforcement learning caused by repeated exposure to identical samples. The name *TikZ30K* refers specifically to the SFT subset, which consists of 29,859 samples. Only 141 samples from the original 30000 samples are discarded during the code augmentation. Please refer to **A4** for detailed clarification about the optimization and post-verification steps.
>
> Moreover, we want to elaborate on our efforts to improve the soundness of the data curation process and final data release.
>  - We initially source **366,075** runnable TikZ snippets. Then, we further identify recurring patterns of low-quality samples to derive a set of filtering rules, followed by employing Qwen-2.5-VL-32B  to automatically assign 5-point quality scores to each image, yielding the final **225,648** high-quality samples.
> - Given that our initial objective is creating a high-quality dataset for efficient cold-start training, we do not apply the expensive LLM-based reordering and comment annotation process to all **225,648** data. We perform stratified sampling based on token length to select a representative subset of **58K** samples. This subset is divided into **30K** samples for SFT and 28K for RL.
> - We have also recognized that different research purposes may have varying requirements for data size. To accommodate this diversity, we have applied the LLM-based reordering and comment annotation optimization to the entire **225,648** samples. This complete, high-quality version will be publicly released alongside the smaller 30K subset. Researchers seeking larger-scale training data or alternative sampling strategies can thus access the fully curated corpus and adapt it to their specific needs, **eliminating the need for additional financial costs associated with optimizing data with the LLM APIs**. Those prioritizing efficiency can directly use the compact 30K version for rapid prototyping and cold-start experiments.
>
> We have revised both Appendix C.4 and Section 3.2 to clarify the data filtering process.

---

> ### Author Response · Authors · 2025-11-21
> **Response to Reviewer modw (4/4)**
>
> **Q4**: The code augmentation process does not seem trivial for an LLM. The paper mentions that the procedure was repeated when rendering failed or when the rendered image was dissimilar to the original. On average, how many attempts were required per sample to obtain a valid code conversion? Moreover, were all samples successfully converted into valid code, or were some data filtered out due to unsuccessful conversions?
>
> **A4**: We initially also shared the intuition that the code augmentation process is challenging for an LLM. However, we found that the LLM could properly recognize the inherent visual structure represented by the code and reorder it. We attribute the *unexpected* capability to the semantic clarity of the TikZ grammar and the LLM's powerful code understanding capability.
> In practice, the actual challenge for an LLM in reordering is correctly updating variable and coordinate dependencies since reordering would change the code order.  Subtle false dependency adjustments may lead to compilation failures. Therefore, we employ a refinement strategy: the model is repeatedly prompted to correct the code using the first error extracted from the error logs as feedback, up to three rounds.
>
> Taking the TikZ30K as an example:
> - In the first round, 9 out of 30,000 samples (0.03%) failed due to the LLM reply not following the format, leading to code extraction errors. - Among the remaining 29,991 extracted samples, 1,656 (5.52%) either failed to compile or produced rendered images dissimilar to the original, using a normalized MSE threshold of 60 for detecting dissimilarity. This threshold accounts for minor visual variations, as the rerendered image, rather than the original, is used as the training target, making small deviations acceptable.
> - These 1,665 (5.55%) problematic cases (9 + 1,656) were passed into the second refinement round, where the codes of all samples were successfully extracted, 1150 (3.83%) of which compiled successfully and passed the similarity check.
> - The remaining 515 (1.72%) samples are passed to the third round, 374 passed.
> - Ultimately, 141 samples (0.47% of the original dataset) could not be resolved and were filtered out.
>
> In summary, after three rounds of iterative refinement, the overall success rate improved from 94.45% to 99.53%, yielding a final high-quality dataset of 29,859 reliably compiled and visually consistent TikZ samples.
>
> We have carefully revised the *Sect. C.4 Code Augmentation*  for clarity. Thank you for bringing the missed details to our attention!
>
> ----------------------
>
> **Q5**: Could you please clarify which specific tools or libraries were used to extract the elements from the PDF files when computing the geometric similarity reward?
>
> **A5**:  We use the Python library `PyMuPDF` to extract both textual and geometric elements from PDF files. Specifically:
> - It first parses the PDF page via `page = pymupdf.open(filename)[0]`;
> - For text content, we access a dictionary using `text_dict = page.get_text("dict")`, where the key is the text content and the value is the coordinates;
> - For geometric elements, we retrieve drawing commands via `page.get_drawings()`. Each drawing item includes an `item_type` field that indicates its type. For example, `re` for rectangle and `l` for line. The drawing object also contains the specific drawing commands that could be further parsed.
> - Also, we want to clarify that this extraction process is based on the excellent properties of TikZ-generated PDFs, which embed text and geometric primitives as clean vector elements, preserving their structure.
>
> ----------------------
>
> **Q6**: In the computation of the geometric similarity reward, I assume that matching the corresponding elements is not a trivial task. Could you please elaborate on how reliable the matching results are when inspected by humans? In other words, to what extent does the proposed algorithm produce visually or semantically reasonable correspondences between elements?
>
> **A6**:  Thank you for your insightful question. We elaborate on this by providing visualizations of the match results between geometric elements.  Specifically, we have added
> - Figure 18 (Appendix E.2) illustrates geometric element matches, covering a spectrum of cases from simple primitives to complex composite structures, and additional
> - Figure 17 (Appendix E.1) shows correspondences between text elements, inspired by your insightful question on geometric element matches.
>
> The results show that our proposed algorithm could produce visually or semantically reasonable correspondences between elements.

---

> ### Author Response · Authors · 2025-11-25
> **Official Comment by Authors**
>
> Dear Reviewer modw,
>
> I hope this message finds you well.
>
> As the discussion period is ongoing and time is running short, we wanted to ensure we have addressed all your concerns satisfactorily. If there are any additional points or feedback you'd like us to consider, please let us know. Your insights are invaluable to us, and we're eager to address any remaining issues to improve our work.
>
> Thank you for your time and effort in reviewing our paper.
>
> Best regards,
>
> Authors

---

### Author Response · Authors · 2025-12-02
**Summary of Revisions**

Dear Reviewers and ACs,

We sincerely thank you for your time and valuable feedback during the review and rebuttal period.

We are grateful that reviewers recognized the strengths of our work, including:
**(1) the well-motivated and innovative data optimization pipeline with solid ablation studies** (Reviewers modw, 1soJ, jdjw, and zdpU),
**(2) the comprehensive and effective reward design** (Reviewers modw, 1soJ, and zdpU),
**(3) the insightful experimental analysis and findings** (Reviewers 1soJ and zdpU), and
**(4) the well-written presentation** (Reviewers jdjw and zdpU).

During the rebuttal period, we have meticulously addressed each of the reviewers' suggestions and conducted additional experiments and clarifications to further support our conclusions.  Modifications have been incorporated into the revised manuscript, with changes highlighted in **blue**. The reviewer feedback and key updates are summarized below:

### **Reviewer Feedback:**
> - **Reviewer modw** did not reply to our response before the leak incident, with an initial score of **6**.
> - **Reviewer 1soJ** (commented on 25 Nov 2025, 11:54 UTC, before the leak incident): appreciated our rebuttal, **decided to maintain the positive score 6**, and asked three follow-up questions.
> - **Reviewer jdjw**  (commented on 25 Nov 2025, 18:04 UTC, before the leak incident): Score improved from **4** to **6**. The reviewer noted that "It addressed all of my concerns, and I have raised my score accordingly".
>- **Reviewer zdpU** (commented on 25 Nov 2025, 18:25 UTC, before the leak incident): appreciated our rebuttal and additional experiments, and **decided to maintain the positive score of 6**.
>- We regret that, due to the OpenReview information leakage incident, we were unable to further discuss our replies to Reviewer modw’s first-round review and Reviewer 1soJ’s follow-up questions. We believe our replies and revisions adequately address their concerns.

### **Additional Experiments and Analysis:**
> - **Human Evaluation**: We conducted comprehensive human studies to compare our model with the leading open-sourced and proprietary models identified in the automatic evaluation separately. (Reviewer modw W2, Reviewer 1soJ W3, **Sect. 4.4**)
> - **Code Efficiency Metric**: We added token and character length metrics for measuring code efficiency. (Reviewer 1soJ W1, **Appendix D.3**)
> - **Additional Automatic Evaluation**: We evaluated another test set consisting of 2000 samples. (Reviewer jdjw W3, **Appendix D.4.2**)
> - **Additional Data Ablation Study**: We ran our data optimization pipeline on DatTikZ-V3-Public and validated its effectiveness. (Reviewer zdpU W1, **Appendix D.6**)
> - **Visualization of Textual/Geometric Element Matching Results**: We added comprehensive visualizations presenting the matching results between extracted and ground-truth textual/geometric elements. (Reviewer modw Q6, Reviewer jdjw W4, **Appendix E.1 Figure 17 and E.2 Figure 18**)
> - **Statistical Analysis over Each Data Filtering Step**: We provided a step-wise breakdown of how the collected data were progressively refined into the final TikZ30K, accompanied by statistical analysis at each stage. (Reviewer jdjw W2, **Appendix C.2**)


### **Clairification:**
> - **Dataset Licenses**: We included complete license information for all data sources used in our collection. (Reviewer modw W1, **End of the Main Content**)
> - **Data Filtering Process**: We revised the main content to be self-contained, with a clear explanation of how TikZ30K was derived from the initially collected data. (Reviewer modw Q3, Reviewer jdjw W1, Reviewer zdpU W2, **Section 3.2 and Appendix C.2**)
> - **Dataset Timeframe**:  We added the timeframe of the collected data and emphasized the strict temporal separation between the training and test sets.  (Reviewer 1soJ W2, **Section 3.2**)
> - **Design Choice of Freezing ViT & MLP during SFT**: We added a loss comparison figure contrasting the SFT losses when freezing versus unfreezing the ViT and MLP components. (Reviewer modw Q1, **Appendix E.3 Figure 19**)
> - **Code Augmentation Process**: We included detailed success rates for each round of optimization. (Reviewer modw Q4, **Appendix C.4**)
> - **Uneven spaced token length-based sampling**: We clarified that our stratified sampling strategy uses unevenly spaced bins, determined by analyzing the token length distribution of the initial dataset. (Reviewer jdjw W1, **Appendix C.4**)
> - **Library for Extracting Elements from PDFs**: We specified that we use the Python library PyMuPDF for extracting vector elements from PDFs. (Reviewer modw Q5, **line 272**)

Thank you once again to all the reviewers for your constructive feedback, which has helped us enhance our work to make it more solid and comprehensive. We believe these extensive experiments and revisions have significantly strengthened our paper and thoroughly addressed the concerns raised.

Best regards,

All Authors

---

### Meta-Review · Area_Chair_ggr8 · 2026-01-07

**Summary:**

The paper was generally viewed as addressing an important and timely problem: Generalized Scientific Diagram Parsing.

The authors propose a two-stage SFT+RL framework and a carefully designed data curation pipeline (based on high-quality diagram-TikZ code pairs).

However, concerns were raised regarding dataset transparency and size, potential evaluation bias, lack of human evaluation, and clarity of technical claims.

**Reviewer Concerns:**

The rebuttal substantially addressed most of these issues, improving the paper’s overall credibility.

**Reviewer Scores:**

The reviewers would recommend acceptance for this paper after rebuttal. AC also holds positive recommendation for this paper.

---

### Decision · Program_Chairs · 2026-01-26

Accept (Poster)